# LIME: LESS IS MORE FOR MLLM EVALUATION

## ABSTRACT

Multimodal Large Language Models (MLLMs) are measured on numerous benchmarks like image captioning, visual question answer, and reasoning. However, these benchmarks often include overly simple or uninformative samples, making it difficult to effectively distinguish the performance of different MLLMs. Additionally, evaluating models across many benchmarks creates a significant computational burden. To address these issues, we propose LIME (Less Is More for MLLM Evaluation), a refined and efficient benchmark curated using a semi-automated pipeline. This pipeline filters out uninformative samples and eliminates answer leakage by focusing on tasks that require image-based understanding. Our experiments show that LIME reduces the number of samples by 76% and evaluation time by 77%, while it can more effectively distinguish different models' abilities. Notably, we find that traditional automatic metrics like CIDEr are insufficient for evaluating MLLMs' captioning performance, and excluding the caption task score yields a more accurate reflection of overall model performance. All code and data are available at `https://anonymous.4open.science/r/LIME-49CD`.

## 1 INTRODUCTION

In order to better understand the model's capabilities and guide addressing the shortcomings of MLLMs, researchers develop numerous benchmarks for various tasks (Antol et al., 2015; Wei et al., 2023; Fu et al., 2023; Yue et al., 2024; Wu et al., 2024a). These benchmarks thoroughly explore the capabilities of MLLMs in various tasks such as image captioning, image question answering, and multimodal retrieving.

However, existing MLLM benchmarks and unified evaluation frameworks cannot effectively and efficiently reflect the ability of MLLMs. Current benchmarks include numerous relatively simple samples (i.e., how many chairs are in the picture) and some incorrect questions caused by annotation issues. Most MLLMs consistently perform on these samples (i.e., all correct or all wrong). Therefore, those benchmarks cannot fully reflect the gap between different MLLMs and across various tasks. Besides, the current unified multimodal evaluation frameworks require significant computational resources, necessitating integrating much evaluation data from various benchmarks. The selection of effective evaluation data is largely overlooked by current researchers.

As shown in Figure 1, to address the aforementioned issues, we propose to use a general data process pipeline and curate a LIME, which contains 9403 samples and is refined across 10 tasks within 6 domains. We select six major tasks in the multimodal domain and use 9 MLLMs to refine those 10 benchmarks within the corresponding domain. To eliminate bias introduced by individual models, we choose 9 models as judges and filter samples based on their performance. On the one hand, we remove samples that most models answer correctly due to the fact that they cannot distinguish the capabilities among different models. On the other hand, we use a method that combines humans and MLLMs to filter out some abnormally difficult samples. Meanwhile, we use LLMs to filter out samples that can be answered directly from the context of the question. After that, we obtain a smaller yet higher-quality unified bench (i.e., LIME).

We conduct various experiments on LIME using both MLLMs and LLMs on different input settings, such as QA + image inputs, QA input (text-only input), and the QA + image description experiment. We make several valuable findings:

- LIME can better reflect the performance differences of MLLMs. On our LIME benchmark, under consistent conditions (same model series, same model size), different MLLMs demonstrate a wider score range, indicating that LIME is more effective at reflecting performance differences between models with a smaller amount of data.

- MLLMs exhibit varying capabilities across different subtasks. Specifically, they excel in the Visual Question Answering (VQA) subtasks, showcasing relatively high performance when answering questions directly related to factual information depicted in images. However, their performance is comparatively lower in tasks that necessitate the application of additional commonsense knowledge or complex reasoning. This highlights the significant image content recognition capabilities of current MLLMs.

- Through the correlation analysis of scores across different tasks, we find that using traditional automatic metrics for the captioning task makes it difficult to reasonably evaluate the model's performance. Different tasks have varying requirements for factual perception and the application of additional commonsense knowledge in images.

## 2  METHOD

**Most benchmarks contain low-quality, noisy data.** Figure 2 shows the statistics of different subtasks within our LIME benchmark. It is worth mentioning that the proportion of easy and wrong samples exceeds 30Out of the 10 subtasks, 6 have proportions exceeding 50%. Notably, for the POPE dataset, 95% of the data can be classified as noisy or erroneous. This indicates that existing benchmarks are filled with a large amount of low-quality data, which does not accurately reflect the true capabilities of MLLMs.

Inspired by MMStar (Chen et al., 2024a), we utilize open-source MLLMs and LLMs as the judges for filtering, specifically, we remove the existing annotation errors. The overall pipeline consists of three main stages: (1) Using open-source models as judges, (2) A semi-automated screening process, and (3) Eliminating answer leakage. Our approach aims to improve existing benchmarks by removing inaccurate and oversimplified data.

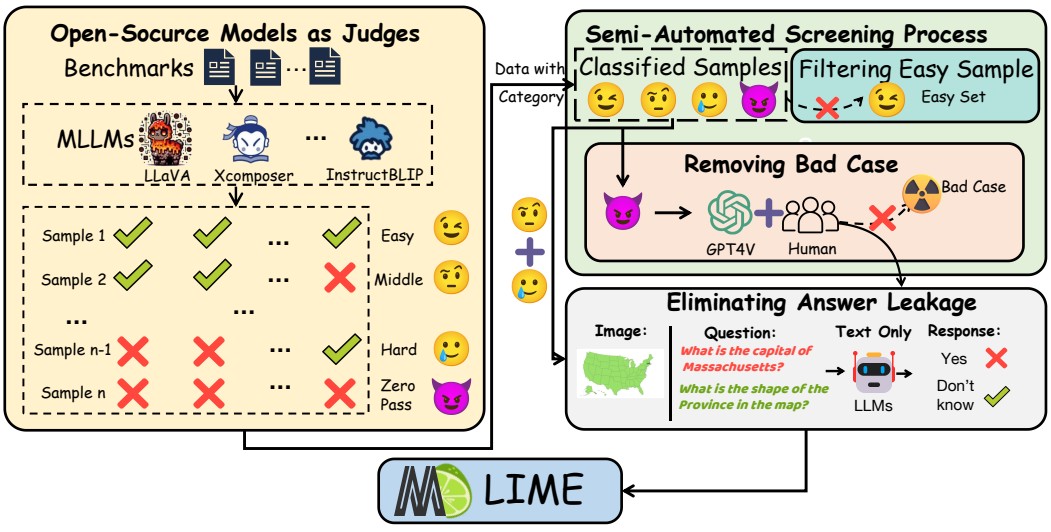

Figure 1: Pipeline of the Data Curation. The left half part is the Open-Source Models as Judges module, which uses several Multimodal LLMs to answer questions for each sample and assess their difficulty. The upper right part is the Semi-Automated Screening Process module filtering some samples that are too simple or difficult. As for the Eliminating Answer Leakage, we filter the sample that can be answered without the image.

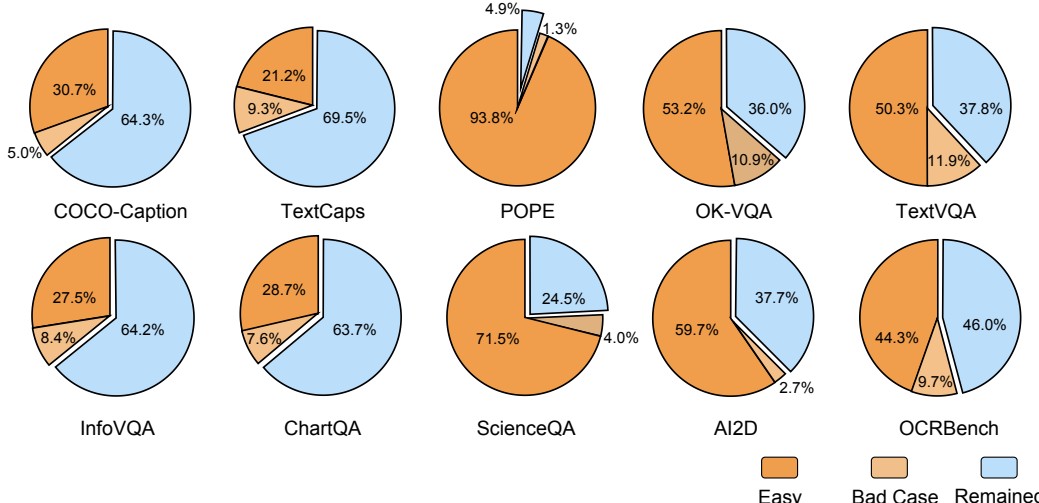

Figure 2: Overall data statics about selected subtasks. **Easy:** questions that most models can answer correctly, **Bad Case:** questions that may contain errors, **Remained:** questions that finally remain.

### 2.1 OPEN-SOURCE MODELS AS JUDGES

To avoid potential biases that may exist in individual MLLMs, we select ten different types of open-source models as judges. To categorize the difficulty of each sample, we analyze the performance of all judge models on each question and label the difficulty based on the number of models that answer correctly. We define $N$ as the number of models that correctly answer the sample. If $N \geq 6$, the question is classified as the easy set. If $3 \leq N \leq 5$, it is classified as the middle set. Conversely, if $N \leq 2$, it is classified as the hard set.

### 2.2 SEMI-AUTOMATED SCREENING PROCESS

Easy samples do not effectively differentiate the capabilities of various models, as most models can answer them correctly. Therefore, we remove the easy samples to assess model performance better.

Furthermore, we find that some questions are not correctly answered by any model, which can be due to potential errors in the question design. To mitigate these potential errors and filter out totally incorrect questions, we implement a semi-automated screening process, which consists of two stages. In the first stage, all questions with zero passes are reviewed by GPT-4V to assess their correctness in terms of logic and meaning. In the second stage, questions deemed correct by GPT-4V are then manually screened. This strategy helps us eliminate meaningless or erroneous data from the dataset, thereby reducing its size and improving its quality.

### 2.3 ELIMINATING ANSWER LEAKAGE

Although the previous two stages have filtered out potential errors and assessed the quality of the questions, we still need to address the potential issue of **ANSWER LEAKAGE**. Multimodal Answer Leakage can be summarized into two main categories: 1.*Text Answerable Questions*: The textual information contains all the necessary details to answer the question, making the corresponding visual information redundant. 2.*Seen Questions*: The MLLMs have encountered a specific question during training and has memorized the question along with its corresponding ground truth.

As for the **Seen Questions**, it has been removed in the Filtering Easy Sample module in Sec. 2.2. Therefore, we conduct a text-only check using pure text LLMs to eliminate the **ANSWER LEAK-AGE**. Specifically, based on LLMs' responses, we remove the samples that can be directly answered without using the image. After that, we proportionally sample 1,200 samples from these categories based on their difficulty levels. For benchmarks with fewer than 1,200 entries, we adapt all samples.

## 3 LIME: A COMPREHENSIVE MLLMS BENCHMARK

In this section, we propose LIME, a comprehensive benchmark for Multimodal Large Language Models (MLLMs). LIME streamlines existing mainstream benchmarks. Tab 1 shows the main datasets included in our benchmark, as well as the data scale after careful pruning. For each sub-dataset, we aim to keep the size around 1k samples.

Table 1: Data statics: **Full Size:** the size of the original dataset, **Lite Size:** the final size of the LIME. For the COCO-Caption dataset, we selected the 2017 subset, and for the ScienceQA dataset, we chose the ScienceQA-IMG subset.

| Task Domain | Dataset | Split | Full Size | Lite Size |
|---|---|---|---|---|
| Captioning | TextCaps | val | 3166 | 1200 |
| | COCO-Caption | val | 5000 | 1200 |
| T/F reasoning | POPE | val | 9000 | 443 |
| Normal VQA | OK-VQA | val | 5046 | 1200 |
| | TextVQA | val | 5000 | 1200 |
| Infographic QA | infoVQA | val | 2801 | 1200 |
| | ChartQA | val | 2500 | 1200 |
| Science QA | ScienceQA | val | 2097 | 300 |
| | AI2D | val | 3088 | 1000 |
| OCR | OCRBench | val | 1000 | 460 |

### 3.1 TASK DEFINITION

We have categorized the existing mainstream tasks into six domains: Captioning, T/F Reasoning, Normal VQA, Infographic Understanding QA, Science QA, and OCR. Below are the task definitions for each domain

**Image understanding and captioning**: The Captioning task focuses on the fundamental image-text understanding ability, requiring MLLMs to accurately describe and understand the content of images. This ability is commonly learned by most multimodal models during the pre-training stage. For example, the CLIP model aligns image and text features through contrastive learning, making Captioning a measure of the basic capabilities of MLLMs.

**T/F reasoning**: T/F Reasoning requires the model to judge the truthfulness of textual statements based on the image content. This not only demands basic image understanding from the MLLMs but also requires a certain level of reasoning ability.

**Normal VQA**: Normal VQA, or Visual Question Answering, comprehensively evaluates the model's ability to answer questions based on visual input. MLLMs are required to select the most appropriate answer from specific options.

**Infographic Understanding QA**: This task differs from Normal VQA as it tests the model's ability to retrieve details from images. MLLMs need to find the most relevant information in the image related to the question and then provide a reasoned answer.

**Science QA**: Science QA includes questions and answers related to natural science knowledge. This requires the model to have domain-specific knowledge in natural sciences, mainly assessing the MLLMs' mastery of knowledge within a specific domain.

**OCR**: The OCR task requires the precise extraction of textual content from images.

### 3.2 DATA STATISTICS

LIME is composed of 10 open-source multimodal evaluation benchmarks, with scales ranging from 1,000 to 9,000. After our three-stage data curation, the data scale of each benchmark is significantly reduced. Figure 1 shows the number of samples removed at each stage compared to the original dataset. The amount of data removed varies at each stage, with the most being removed in the first stage, reflecting a large number of low-difficulty or data-leakage samples in the existing 9 MLLMs. Comparing the data volumes before and after the second stage of semi-automated screening, we can see that many datasets, such as OK-VQA and TextVQA, have a high rate of low-quality data leading to MLLMs' incorrect answers. Additionally, some datasets, such as ScienceQA and AI2D, have a significant amount of data removed after the third stage, indicating that many questions in these datasets may contain potential answer leakage. The statistics of the curated data are shown in Tab 1.

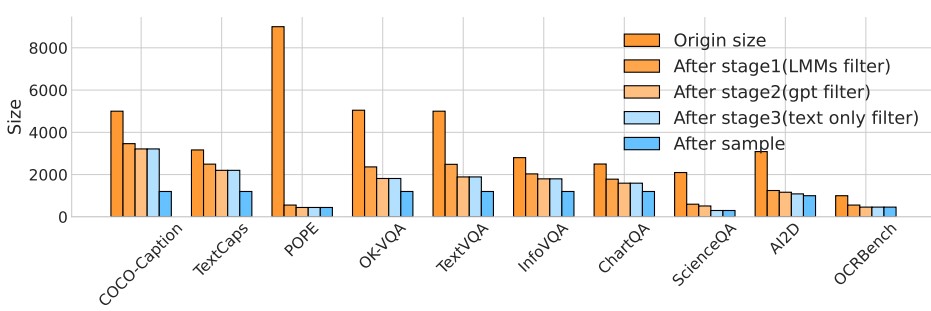

Figure 3: The number of samples removed at each stage compared to the original data, including three stages of filtering and the final sampling stage.

# 4 EXPERIMENT

## 4.1 EXPERIMENT SETTING

To evaluate the quality of LIME, we conduct a series of experiments across various open-source and closed-source models. These experiments primarily encompass the following three settings:

**Main experiment**: To demonstrate the performance of LIME, we evaluate mainstream open-source and closed-source models using a standardized process to reflect their overall performance differences.

**Text-only set**: To prevent potential data leakage issues, we conduct validation experiments using text-only QA pairs. This verifies whether LLMs can correctly answer questions based on text-only information.

**Text-only question with Image Description set**: Image Description (ID) refers to simple descriptions of images that represent superficial information contained within them. For most MLLMs, questions containing only superficial information are easy to answer; however, questions requiring complex visual inference are significantly more challenging. To further validate whether LIME can reflect the capabilities of MLLMs, we input text-only QA pairs combined with ID into LLMs and test their ability.

## 4.2 BASELINES

We select LLaVA-1.5 (Liu et al., 2023a;b), LLaVA-1.6 (Liu et al., 2024), Tinny-LLaVA (Zhou et al., 2024), MiniCPM (Hu et al., 2024), Idefics-2 [1], Deepseek-VL(Lu et al., 2024), CogVLM (Wang et al., 2023; Hong et al., 2023), XComposer-4KHD (Zhang et al., 2023), Mantis (Jiang et al., 2024), InternVL-1.5 and InternVL-2 (Chen et al., 2023; 2024b) as our MLLMs baseline, and LLaMA3, Yi, Yi-1.5 (AI et al., 2024), Qwen-1.5 (Bai et al., 2023a) and Qwen2 (Yang et al., 2024) as LLMs baseline. To ensure fairness in the evaluations, we use the unified evaluation framework provided by lmms-eval (Zhang et al., 2024b) to conduct evaluation experiments on LIME. For models not supported by lmms-eval, we refine the inference code provided by the model developers to make it compatible with the new models for the sake of aligning the results of different models.

**Metrics** For most tasks included in LIME, we reference the metrics computation methods used in lmms-eval. Specifically, for tasks such as AI2D, ScienceQA, OCRBench, and POPE, we calculate the accuracy of the extracted responses. For tasks such as OK-VQA and TextVQA, we calculate the metric scores based on the overlap between the response and the candidate answers. For tasks like TextCaps and COCO-Caption2017, we use CIDEr as the score. The ANLS metric is used for the infoVQA task, and the Relaxed Overall metric is used for the ChartQA task.

We calculate the sub-scores for each task category by taking a weighted average of the subtask scores, and then compute the overall score by weighted averaging the scores of all tasks except for the caption tasks. The details of the metrics calculation are provided in Tab 7.

---

[1] https://huggingface.co/blog/idefics2

# 5 RESULTS

## 5.1 MAIN RESULT

Table 2: **Left half of the table:**Comparing overall scores of LIME and Original. The arrow next to the LIME score indicates the change in ranking on LIME compared to the original dataset. ↑: upward shift, ↓: downward shift, and -: no change. **Right half of the table:** performance on six domains

| Model | Size | LIME | Original | Reasoning | VQA | InfoQA | SciQA | OCR | Caption |
|---|---|---|---|---|---|---|---|---|---|
| GPT-4O | - | **52.63** | - | 47.18 | 42.95 | 57.63 | 56.15 | 72.39 | 47.84 |
| claude-3-5-sonnet | - | 51.99 | - | 35.89 | 50.33 | 56.38 | 44.69 | 73.91 | 28.00 |
| Gemini-1.5-Pro-Vision | - | 49.46 | - | 54.63 | 37.71 | 55.33 | 50.15 | 73.26 | 41.38 |
| GPT-4-Vision-Preview | - | 42.23 | - | 42.44 | 33.86 | 48.00 | 42.39 | 55.22 | 29.14 |
| InternVL-2 2023 | 40B | 66.85 ( - ) | 80.31 | 51.69 | 48.72 | 81.12 | 75.92 | 75.87 | 56.02 |
| Qwen2-VL 2023b | 7B | 65.28 (↑ 1) | 79.14 | 53.05 | 51.37 | 80.83 | 62.08 | 77.61 | 89.67 |
| InternVL-1.5 2024b | 26B | 64.12 (↓ 1) | 79.49 | 51.69 | 52.68 | 78.96 | 63.32 | 60.65 | 90.93 |
| InternVL-2 2023 | 26B | 63.98 ( - ) | 78.82 | 54.63 | 45.64 | 79.12 | 70.54 | 71.09 | 66.54 |
| InternVL-2 2023 | 8B | 62.00 (↑ 1 ) | 77.84 | 49.21 | 45.15 | 76.00 | 68.54 | 70.65 | 34.00 |
| LLaVA-OneVision 2024 | 7B | 61.95 (↓ 1 ) | 78.71 | 52.37 | 51.27 | 74.50 | 66.77 | 47.83 | 106.46 |
| XComposer2-4KHD 2023 | 7B | 57.52 (↑ 4) | 71.93 | 46.28 | 44.22 | 73.29 | 58.38 | 53.04 | 87.57 |
| InternVL-2 2023 | 4B | 57.22 (↓ 1) | 73.97 | 47.18 | 39.89 | 71.21 | 63.31 | 67.17 | 28.83 |
| CogVLM-2 2024 | 19B | 54.44 (↑ 6) | 69.93 | 51.02 | 37.19 | 69.92 | 54.00 | 68.26 | 28.84 |
| Qwen2-VL 2023b | 2B | 54.00 (↑ 5) | 70.86 | 50.79 | 43.78 | 66.25 | 46.38 | 68.04 | 88.39 |
| InternVL-2 2023 | 2B | 53.64 (↓ 2) | 73.00 | 50.79 | 40.71 | 62.88 | 56.54 | 67.39 | 47.27 |
| CogVLM-1 2023 | 17B | 51.03 (↑ 1) | 71.34 | 55.10 | 51.45 | 59.46 | 36.54 | 41.96 | 33.92 |
| Cambrian 2024 | 34B | 50.17 (↓ 5) | 73.26 | 49.44 | 39.66 | 57.50 | 60.23 | 39.13 | 4.62 |
| Cambrian 2024 | 13B | 48.57 (↓ 4) | 72.39 | 50.79 | 41.53 | 56.04 | 49.23 | 42.39 | 6.96 |
| InternVL-2 2023 | 1B | 48.21 (↑ 3) | 68.46 | 52.82 | 36.46 | 56.04 | 47.92 | 65.00 | 14.19 |
| Cambrian 2024 | 8B | 47.95 (↓ 4) | 71.84 | 49.89 | 42.12 | 53.55 | 49.46 | 43.04 | 6.13 |
| LLaVA-1.6 2024 | 34B | 44.06 (↑ 3) | 67.22 | 47.00 | 30.80 | 53.21 | 53.08 | 37.17 | 66.25 |
| MiniCPM-LLaMA3-2.5 2024 | 8B | 42.61 (↓ 3) | 71.22 | 43.10 | 43.55 | 58.55 | 6.60 | 55.87 | 35.89 |
| LLaVA-OneVision 2024 | 0.5B | 41.40 (↑ 4) | 65.65 | 48.98 | 35.87 | 48.04 | 36.23 | 42.83 | 93.34 |
| LLaVA-LLaMA3 2023 | 8B | 40.90 (↓ 3) | 69.74 | 44.24 | 37.36 | 43.33 | 45.56 | 30.22 | 74.03 |
| Mantis-Idefics-2 2024 | 8B | 39.25 ( - ) | 66.91 | 44.24 | 36.79 | 39.75 | 43.69 | 32.17 | 82.44 |
| Deepseek-VL 2024 | 7B | 38.10 (↑ 2) | 65.62 | 48.50 | 34.90 | 38.50 | 44.23 | 25.43 | 68.72 |
| LLaVA-1.6-vicuna 2024 | 13B | 37.08 (↓ 4) | 67.29 | 43.10 | 30.00 | 41.63 | 41.54 | 31.96 | 62.23 |
| Idefics-2 2024 | 8B | 36.39 (↓ 2) | 66.73 | 42.00 | 46.05 | 18.50 | 47.46 | 42.61 | 77.87 |
| LLaVA-1.6-vicuna 2024 | 7B | 30.15 ( - ) | 64.80 | 41.10 | 25.75 | 32.88 | 31.77 | 23.70 | 62.20 |
| Mantis-SigLIP 2024 | 8B | 29.13 (↑ 1) | 58.96 | 45.60 | 29.39 | 25.79 | 35.77 | 10.65 | 74.69 |
| MiniCPM 2024 | 1.0 | 26.15 (↑ 2) | 56.18 | 44.00 | 21.60 | 24.58 | 35.46 | 14.57 | 72.80 |
| LLaVA-1.5 2023a | 13B | 20.38 (↓ 2) | 59.58 | 36.60 | 25.80 | 8.96 | 31.08 | 5.87 | 74.81 |
| LLaVA-1.5 2023a | 7B | 17.20 (↓ 1) | 57.27 | 32.51 | 19.97 | 7.17 | 29.81 | 4.78 | 72.47 |
| InstructBLIP-vicuna 2023 | 7B | 15.55 ( - ) | 47.87 | 45.10 | 16.75 | 6.04 | 24.77 | 4.35 | 77.61 |
| Tiny-LLaVA-1 2024 | 1.4B | 13.95 ( - ) | 34.30 | 37.00 | 9.80 | 8.33 | 27.85 | 3.48 | 61.05 |

As shown in Tab 2, we evaluate both open-source and closed-source MLLMs using our LIME benchmark. Overall, for closed-source models, GPT-4O achieves the best performance with a score of 52%, while for open-source models, models with larger parameter sizes and newer versions tend to have higher overall scores. InternVL-1.5, InternVL-2-Large (26B, 40B), and LLaVA-OneVision-7B achieve the best overall performance, with their overall scores all surpassing 60%. The performance of InternVL-2-Small (1B-8B), the CogVLM series, and the Cambrian series follows, with their overall scores ranging from 45% to 60%.

Comparing the overall scores of LIME and Origin benchmarks, we observe that certain model series, such as Cambrian and LLaVA-1.5, experience a decline in overall scores. Conversely, the CogVLM and LLaVA-OneVision series show an improvement, with CogVLM2 and XComposer-4KHD experiencing significant increases of 4% and 6%, respectively.

Tab 6 provides more detailed experimental results. Regarding caption subtasks, most models demonstrate good performance. These tasks involve generating or assessing descriptions of the content in images, which indicates that current MLLMs possess strong image content recognition capabilities. As for the VQA task, current MLLMs perform relatively well on TextVQA, ChatQA, and ScienceQA, where the questions directly ask about facts in the picture. However, their performance is relatively lower on OK-VQA, infoVQA, and AI2D, which require additional commonsense knowledge or complex reasoning to answer the questions. This demonstrates that current MLLMs exhibit significant image content recognition capabilities but are limited in their ability to perform complex reasoning using additional knowledge. We believe this limitation may be due to constraints in the language model component of MLLMs.

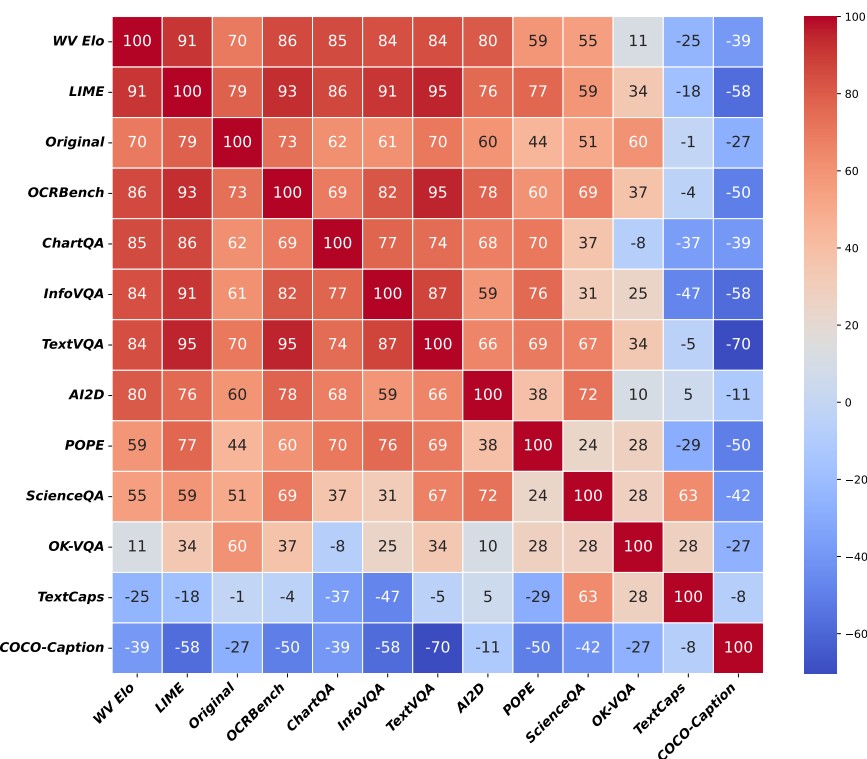

Figure 4: Correlation distribution between LIME and Wildvison Elo.

## 5.2 CORRELATION ANALYSIS

Figure 4 illustrates the correlation between the various sub-tasks in LIME and WildVision Bench. Most tasks in LIME exhibit a strong positive correlation with WildVision Bench. Six subtasks have correlation scores exceeding 80%. Additionally, the overall score of LIME correlates at 91% with WV-Elo, which is higher than any individual sub-task and the original bench's correlations, demonstrating that the overall score of LIME provides a more comprehensive reflection of MLLMs' capabilities.

**Automated evaluation metrics (e.g., CIDEr) cannot effectively assess the performance of MLLMs in captioning tasks.** As an early foundational problem, the captioning task is extensively studied, and MLLMs demonstrate exceptional ability in this task. For instance, earlier models like InstructBlip perform exceptionally well on captioning tasks, and there is a broad presence of training data for image captioning in MLLMs' training processes. However, the captioning task shows a negative correlation with all other sub-tasks. This indicates that previous metrics (e.g., BLEU, CIDEr) only focus on the overlap between the model-generated responses and the ground truth, but do not consider that MLLMs might generate content that is semantically close to the ground truth (i.e., the model-generated response may be semantically similar to the ground truth but expressed differently, or the model may generate more detailed content about the image). Consequently, we exclude it from the overall score calculation.

**There is a certain degree of correlation between the sub-tasks in LIME.** On the one hand, the relevance of TextVQA, InfographicVQA, and OCRBench is relatively high. As shown in Fig. 4, the correlation of these tasks all surpasses 85%, and these two VQA tasks require MLLMs to understand fine-grained content in images to answer questions. This demonstrates that OCR tasks also rely on the ability of MLLMs to perceive fine-grained objective facts in images. On the other hand, POPE, ChartQA, and InfographicVQA all require reasoning abilities using extra commonsense knowledge. The correlation scores of these tasks are all above 70%, and POPE requires the model to use extra

knowledge to solve the hallucination of MLLMs. We assume that ChartQA and infoVQA may also necessitate the use of additional common knowledge by the models to solve problems.

## 5.3 EFFECTIVENESS OF LIME

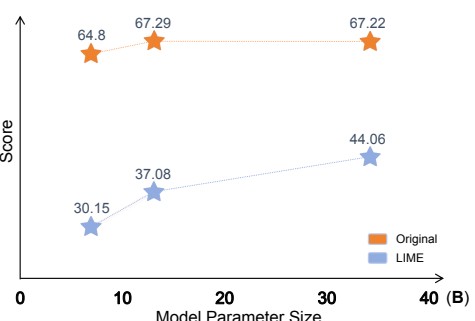 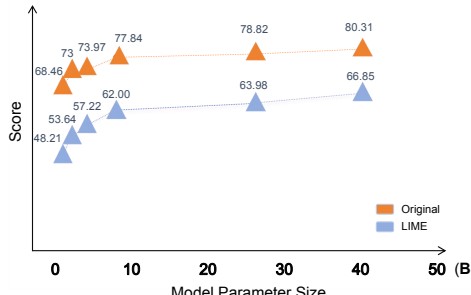

Figure 5: with the same series of models, the distribution differences of various Parameter sizes. Left(★): LLaVA-1.6 series, Right(▲): InternVL-2 series

Table 3: Statistics on the score distributions across different model series.

| Model series | Dataset | GiNi | stdev |
|---|---|---|---|
| InternVL-2 | LIME | 0.061 | 6.972 |
| | Original | 0.030 | 4.421 |
| Cambrian | LIME | 0.006 | 1.227 |
| | Original | 0.002 | 0.715 |
| LLaVA-1.6 | LIME | 0.042 | 6.730 |
| | Original | 0.004 | 1.418 |

Table 4: Statistics on the score distributions across different model sizes.

| Model size | Dataset | GiNi | stdev |
|---|---|---|---|
| 7B | LIME | 0.271 | 19.041 |
| | Original | 0.086 | 10.836 |
| 8B | LIME | 0.128 | 10.685 |
| | Original | 0.046 | 6.270 |
| 13B | LIME | 0.174 | 13.536 |
| | Original | 0.043 | 6.446 |

**LIME provides a more challenging evaluation for MLLMs.** As shown in Tab 2, the MLLMs' performances on LIME are less than those on the Original Bench for most tasks. Compared to the Origin benchmark, different MLLMs show a larger score range on our LIME, indicating that our LIME can better reflect the performance differences between models with a smaller amount of data.

Furthermore, we compare the score variations across different model series and model sizes. Figure 5 illustrates a clear positive correlation between model performance and model size within the same model series. Notably, LIME exhibits a more dispersed score distribution, effectively highlighting the differences in model performance. In Tab 3 and 4, the Gini coefficient and standard deviation are used to measure the differences in overall score distribution across the same model series and model sizes. The larger the Gini coefficient and standard deviation, the greater the disparity in data distribution. It can be observed that, whether within the same model series or the same model size, LIME achieves higher Gini and standard deviation values compared to the original bench. This indicates that LIME can better differentiate the performance differences between various models.

**LIME eliminates potential data leakage.** For multimodal question answering tasks, visual information input is essential, and LLMs are unable to provide correct answers due to they cannot perceive the content within the image. However, as shown in Figure 6 (right), there are severe data leakage issues in the original Bench for the AI2D and ScienceQA tasks. The average score for AI2D is close to 55%, and for ScienceQA, it exceeds 60%, which shows that data from AI2D and ScienceQA in Original are highly likely to have been exposed to the training data of LLMs. In contrast, the LIME has eliminated this potential threat, achieving scores below 25% in AI2D and close to 40% in ScienceQA.

| Model | AI2D | | ScienceQA | |
|---|---|---|---|---|
| | LIME | Original | LIME | Original |
| LLaMA3-8B | 18.10 | 46.76 | 33.33 | 59.35 |
| LLaMA3-70B | 25.70 | 62.05 | 56.00 | 69.91 |
| Qwen1.5-32B | 24.10 | 61.14 | 43.67 | 67.97 |
| Qwen1.5-72B | 19.80 | 57.45 | 35.00 | 61.13 |
| Qwen2-7B | 21.00 | 57.09 | 43.00 | 67.38 |
| Qwen2-72B | 20.60 | 69.95 | 38.67 | 63.36 |
| Yi-1.5-9B-Chat | 20.10 | 23.22 | 17.33 | 23.60 |
| Yi-1.5-34B-Chat | 23.60 | 54.15 | 42.00 | 65.20 |
| Yi-1.5-34B-Chat | 25.20 | 60.69 | 46.00 | 70.55 |

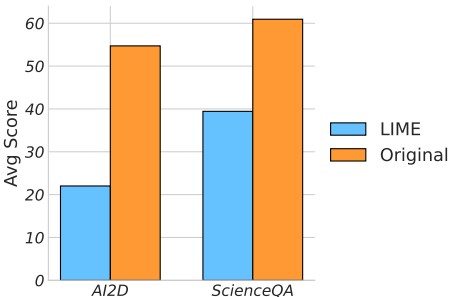

Figure 6: Comparing text only results of LIME and original bench. **Left**: text only results between LIME and Original on AI2D and ScienceQA; **Right**: average score comparison of Original and LIME.

## 5.4 THE IMPACT OF DETAIL IMAGE PERCEPTION

Table 5: Text-only with VD results: With the condition of providing only text QA information and VD information, the performance comparison between vlms-bench and origin bench.

| Setting | Models | AI2D | ChQA | COCO | IVQA | OCRBen | OK VQA | POPE | SciQA | TCaps | TVQA |
|---|---|---|---|---|---|---|---|---|---|---|---|
| LIME | LLaMA3-8B | 23.5 | 6.4 | 2.8 | 12.9 | 9.2 | 17.4 | 32.1 | 16.4 | 5.3 | 17.9 |
| | LLaMA3-70B | 24.0 | 7.7 | 3.3 | 12.3 | 9.3 | 21.8 | 38.1 | 39.4 | 6.0 | 22.0 |
| | Qwen1.5-32B | 28.8 | 6.7 | 6.5 | 9.4 | 8.7 | 4.7 | 39.3 | 46.6 | 9.2 | 13.7 |
| | Qwen1.5-72B | 25.4 | 2.5 | 3.2 | 10.1 | 8.9 | 7.8 | 42.7 | 44.2 | 6.0 | 15.2 |
| | Qwen2-7B | 27.6 | 6.7 | 6.9 | 11.2 | 8.9 | 15.0 | 44.2 | 45.5 | 12.5 | 19.0 |
| | Qwen2-72B | 26.3 | 6.9 | 2.7 | 10.8 | 9.6 | 10.6 | 36.3 | 45.2 | 5.2 | 16.8 |
| | Yi-1.5-9B-Chat | 22.1 | 2.3 | 0.3 | 3.1 | 0.0 | 7.8 | 40.0 | 0.0 | 0.2 | 5.8 |
| Original | LLaMA3-8B | 49.0 | 11.4 | 3.1 | 18.6 | 19.3 | 32.5 | 46.9 | 59.5 | 6.5 | 26.4 |
| | LLaMA3-70B | 52.0 | 12.4 | 3.6 | 17.6 | 19.5 | 36.4 | 5.2 | 64.6 | 7.8 | 36.2 |
| | Qwen1.5-32B | 60.5 | 10.7 | 8.1 | 15.0 | 20.2 | 15.8 | 47.4 | 68.8 | 10.6 | 22.1 |
| | Qwen1.5-72B | 58.8 | 6.4 | 3.8 | 16.6 | 20.2 | 21.1 | 35.1 | 68.4 | 7.1 | 27.4 |
| | Qwen2-7B | 59.2 | 12.7 | 7.4 | 19.7 | 19.6 | 30.5 | 44.6 | 69.0 | 15.4 | 33.3 |
| | Qwen2-72B | 60.4 | 10.5 | 3.5 | 15.7 | 20.5 | 24.2 | 34.3 | 67.9 | 6.8 | 28.7 |
| | Yi-1.5-9B-Chat | 24.7 | 3.1 | 0.0 | 5.9 | 0.5 | 7.8 | 32.7 | 31.7 | 0.2 | 5.8 |

In our data cleaning process, we remove many questions that most models can answer, as well as a small number of questions that are difficult for both humans and GPT-4V to answer, in order to make the benchmark better highlight the differences in model capabilities. As shown in Tab 5, to investigate whether the remaining samples need to be answered by using textual and image information, we conduct experiments using LLMs to generate answers on both the Original Benchmark and MLLMs Benchmark under QID (question + image description) setting.

**LIME requires MLLMs to perceive deeper levels of image information.** Especially in tasks such as AI2D, OCRBench, and TCaps, the scores of LLMs on LIME are significantly lower than on the Original Benchmark when provided with only the questions and simple image descriptions. This indicates that, after removing some of the simpler questions, LIME is better at testing the models' ability to perceive image details.

## 5.5 EXISTING BENCHMARK STILL DIFFERS FROM REAL-WORLD QUERY.

To further investigate the gap between LIME and real-world users' queries, we construct a similarity search system that compares them. MixEval (Ni et al., 2024) uses SentenceTransformers(Reimers, 2019) as the retrieval model, while Uniir (Wei et al., 2023) employs multimodal models like CLIP and BLIP. We use WildVision-Chat as the query data source, which contains 45.2k high-quality user questions, and employ SentenceTransformers to retrieve the top 10 most similar samples from LIME. To fully incorporate image information, we combine the question and image description as the query input. Additionally, we utilize Qwen2-72B to ensure a high level of relevance in the final results. As

a result, we obtain a LIME-fit dataset containing 1.1k relevant samples. **Existing benchmark can't cover all types of real-world query.**

In Figure 9, we compare the category distribution differences between LIME-fit and the WildVision Bench. It is evident that LIME-fit concentrates in a few specific categories (e.g., data analysis, general description, object recognition). However, it does not include instructions for solving real-world problems, such as Face Recognition, Problem Solving, and Scene Description. Furthermore, Figure 10 shows the frequency distribution of each subcategory in LIME-fit, which follows a long-tail distribution. This indicates that the current benchmark does not fully cover the instruction requirements of real-world scenarios.

## 6 RELATED WORK

In recent years, there has been increasing attention on establishing evaluation benchmarks to assess the performance of MLLMs in different scenarios to guide the development of MLLMs. Early multimodal evaluation benchmarks primarily focused on single tasks, such as Visual Question Answering (VQA)(Antol et al., 2015; Goyal et al., 2017; Kafle & Kanan, 2017; Singh et al., 2019; Marino et al., 2019), Image Captioning(Agrawal et al., 2019), and Information Retrieval (Wei et al., 2023). As MLLMs develop, simple benchmarks are no longer sufficient to evaluate the versatile capabilities of these models comprehensively, since most MLLMs demonstrate exceptional ability on those benchmarks. Consequently, numerous more difficult and diverse benchmarks have emerged in recent years to assess the capabilities of MLLMs comprehensively. For instance, MMMU (Yue et al., 2024) and CMMMU (Zhang et al., 2024a) are comprehensive benchmark tests for university-level multidisciplinary multimodal understanding and reasoning. MMBench (Liu et al., 2023c) has developed a comprehensive evaluation pipeline that offers fine-grained capability assessment and robust evaluation metrics. MMRA (Wu et al., 2024b) systematically establishes an association relation system among images to assess the multi-image relation mining ability of MLLMs.

However, those benchmarks cannot distinguish the performance gaps among different models excellently, as they still contain some too simple or difficult samples that most models yield the same results on. Furthermore, training datasets across different models may contain the samples of those benchmarks, which results in data leakage issues (Fu et al., 2023). Mmstar (Chen et al., 2024a) and MMLU_Redux (Gema et al., 2024) have identified several issues within current benchmarks. Mmstar (Chen et al., 2024a) proposes an automated pipeline to filter benchmark data, aiming to detect potential data leakage, while MMLU_Redux (Gema et al., 2024) focuses on correcting annotation errors. However, there is still a pressing need for a comprehensive pipeline that fully addresses the challenges posed by multimodal datasets. In response to this, we introduce LIME: LESS IS MORE FOR MLLM EVALUATION. We have carefully selected six task types from existing mainstream benchmarks and scaled them down according to clear guidelines. This streamlined version retains the core elements of mainstream MLLM benchmarks, providing a more efficient and focused evaluation.

## 7 CONCLUSION

As MLLMs continue to advance, a notable absence of convenient and high-quality multimodal benchmarks has emerged. In response to this, we propose a pipeline aimed at semi-automatically refining existing benchmarks to enhance their quality, culminating in the development of LIME, which comprises 9,403 evaluation samples across 6 types of tasks and 10 different benchmark datasets. By refining the original benchmarks to filter question difficulty and eliminate potentially problematic items, LIME offers a more rigorous evaluation for MLLMs, necessitating a deeper understanding of image information. The outcomes of our evaluation experiments demonstrate the heightened challenge posed by LIME for MLLMs. We anticipate that our approach will contribute to the advancement of MLLM evaluation systems, and we are committed to continually enriching LIME with an expanded array of datasets through regular updates and expansions. Our ultimate goal is to provide the community with a simpler, more efficient, and more accurate evaluation method and suite for MLLMs.

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

# A  APPENDIX

## A.1  OVERALL DATA STATICS

Figure 7shows the overall data distribution in LIME, and figure 8 shows an example for each category title

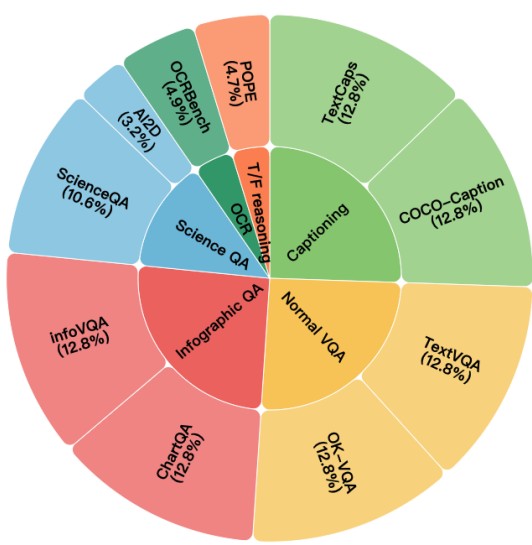

Figure 7: The overall percentage distribution of LIME.

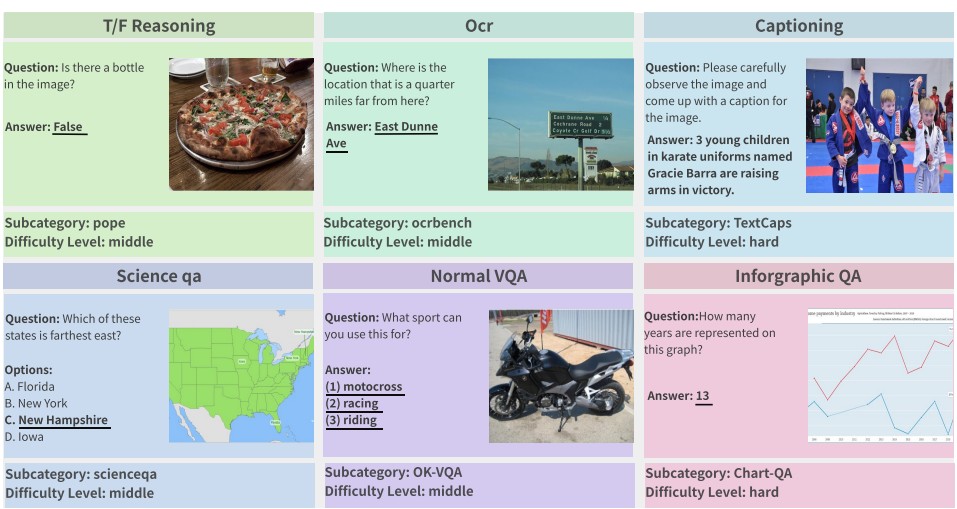

Figure 8: The overview of LIME.

## A.2 MORE EXPERIMENT RESULT

Table 6: Comparing overall scores of LIME and Original. **Top**: results on LIME, **Bottom**: results on the original dataset. The arrow next to the model name indicates the change in ranking on LIME compared to the original dataset. ↑: upward shift, ↓: downward shift, and -: no change.

| Model | Size | Overall | T/F POPE ↑ | Common VQA TVQA ↑ | OK VQA ↑ | InfoVQA ChQA ↑ | IVQA ↑ | ScienceQA AI2D ↑ | SciQA ↑ | OCR OCRBen ↑ | Captioning COCO ↑ | TCaps ↑ |
|---|---|---|---|---|---|---|---|---|---|---|---|---|
| InternVL-2 2023 ( - ) | 40B | 66.85 | 51.69 | 77.98 | 19.45 | 88.33 | 73.92 | 69.20 | 98.33 | 75.87 | 63.10 | 48.94 |
| Qwen2-VL 2023b (↑ 1) | 7B | 65.28 | 53.05 | 74.56 | 28.18 | 83.17 | 78.50 | 58.20 | 75.00 | 77.61 | 68.74 | 110.60 |
| InternVL-1.5 2024b (↓ 1) | 26B | 64.12 | 51.69 | 69.88 | 35.47 | 87.00 | 70.92 | 54.81 | 91.67 | 60.65 | 69.24 | 112.63 |
| InternVL-2 2023 ( - ) | 26B | 63.98 | 54.63 | 75.20 | 16.08 | 87.67 | 70.58 | 62.80 | 96.33 | 71.09 | 76.18 | 56.91 |
| InternVL-2 2023 (↑ 1) | 8B | 62.00 | 49.21 | 66.10 | 24.20 | 83.75 | 68.25 | 60.40 | 95.67 | 70.65 | 42.55 | 25.44 |
| LLaVA-OneVision 2024 (↓ 1) | 7B | 61.95 | 52.37 | 65.22 | 37.32 | 80.83 | 68.17 | 59.20 | 92.00 | 47.83 | 104.74 | 108.18 |
| XComposer2-4KHD 2023 (↑ 4) | 7B | 57.52 | 46.28 | 60.30 | 28.13 | 80.42 | 66.17 | 54.90 | 70.00 | 53.04 | 97.07 | 78.07 |
| InternVL-2 2023(↓ 1) | 4B | 57.22 | 47.18 | 62.29 | 17.48 | 81.92 | 60.50 | 54.00 | 94.33 | 67.17 | 35.99 | 21.67 |
| CogVLM-2 2024 (↑ 6) | 19B | 54.44 | 51.02 | 69.46 | 4.92 | 80.33 | 59.50 | 45.00 | 84.00 | 68.26 | 23.67 | 34.01 |
| Qwen2-VL 2023b (↑ 5) | 2B | 54.00 | 50.79 | 70.70 | 16.87 | 67.50 | 65.00 | 42.90 | 58.00 | 68.04 | 75.06 | 101.72 |
| InternVL-2 2023 (↓ 2) | 2B | 53.64 | 50.79 | 59.56 | 21.87 | 71.75 | 54.00 | 45.80 | 92.33 | 67.39 | 51.95 | 42.59 |
| CogVLM-1 2023 (↑ 1) | 17B | 51.03 | 55.10 | 71.20 | 31.70 | 61.67 | 57.25 | 31.40 | 53.67 | 41.96 | 29.28 | 38.56 |
| Cambrian 2024 (↓ 5) | 34B | 50.17 | 49.44 | 57.28 | 22.03 | 71.83 | 43.17 | 54.80 | 78.33 | 39.13 | 4.27 | 4.97 |
| Cambrian 2024 (↓ 4) | 13B | 48.57 | 50.79 | 58.93 | 24.13 | 69.25 | 42.83 | 45.20 | 62.67 | 42.39 | 7.40 | 6.52 |
| InternVL-2 2023 (↑ 3) | 1B | 48.21 | 52.82 | 57.55 | 15.37 | 65.83 | 46.25 | 37.00 | 84.33 | 65.00 | 15.19 | 13.19 |
| Cambrian 2024 (↓ 4) | 8B | 47.95 | 49.89 | 59.00 | 25.23 | 69.42 | 37.67 | 43.60 | 69.00 | 43.04 | 5.85 | 6.41 |
| LLaVA-1.6 2024 (↑ 3) | 34B | 44.06 | 47.00 | 51.20 | 10.40 | 64.33 | 42.08 | 49.60 | 64.67 | 37.17 | 84.25 | 48.25 |
| MiniCPM-LLaMA3-2.5 2024 (↓ 3) | 8B | 42.61 | 43.10 | 61.80 | 25.30 | 69.00 | 48.10 | 6.60 | 6.60 | 55.87 | 31.91 | 39.86 |
| LLaVA-OneVision 2024 (↑ 4) | 0.5B | 41.40 | 48.98 | 48.61 | 23.13 | 55.42 | 40.67 | 31.20 | 53.00 | 42.83 | 96.40 | 90.28 |
| LLaVA-LLaMA3 2023 (↓ 3) | 8B | 40.90 | 44.24 | 40.01 | 34.72 | 64.42 | 22.25 | 40.72 | 61.67 | 30.22 | 99.35 | 48.71 |
| Mantis-Idefics-2 2024 ( - ) | 8B | 39.25 | 44.24 | 44.51 | 29.07 | 59.00 | 20.50 | 35.80 | 70.00 | 32.17 | 61.82 | 103.07 |
| Deepseek-VL 2024 (↑ 2) | 7B | 38.10 | 48.50 | 44.80 | 25.00 | 54.67 | 22.33 | 37.00 | 68.33 | 25.43 | 54.22 | 83.21 |
| LLaVA-1.6-vicuna 2024 (↓ 4) | 13B | 37.08 | 43.10 | 43.90 | 16.10 | 54.50 | 28.75 | 38.10 | 53.00 | 31.96 | 76.62 | 47.83 |
| Idefics-2 2024 (↓ 2) | 8B | 36.39 | 42.00 | 56.50 | 35.60 | 13.08 | 23.92 | 38.10 | 78.67 | 42.61 | 61.23 | 94.51 |
| LLaVA-1.6-vicuna 2024 ( - ) | 7B | 30.15 | 41.10 | 39.00 | 12.50 | 43.08 | 22.67 | 27.10 | 47.33 | 23.70 | 76.05 | 48.35 |
| Mantis-SigLIP 2024 (↑ 1) | 8B | 29.13 | 45.60 | 26.34 | 32.43 | 35.33 | 16.25 | 27.70 | 62.67 | 10.65 | 68.16 | 81.21 |
| MiniCPM 2024 (↑ 2) | 1.0 | 26.15 | 44.00 | 37.00 | 6.20 | 35.75 | 13.42 | 27.90 | 60.67 | 14.57 | 68.96 | 76.65 |
| LLaVA-1.5 2023a (↓ 2) | 13B | 20.38 | 36.60 | 19.50 | 32.10 | 5.50 | 12.42 | 25.90 | 48.33 | 5.87 | 80.89 | 68.73 |
| LLaVA-1.5 2023a (↓ 1) | 7B | 17.20 | 32.51 | 16.50 | 23.43 | 5.25 | 9.08 | 24.05 | 49.00 | 4.78 | 79.20 | 65.73 |
| InstructBLIP-vicuna 2023 ( - ) | 7B | 15.55 | 45.10 | 11.40 | 22.10 | 3.00 | 9.08 | 21.90 | 34.33 | 4.35 | 102.08 | 53.14 |
| Tiny-LLaVA-1 2024 ( - ) | 1.4B | 13.95 | 37.00 | 18.70 | 0.90 | 4.50 | 12.17 | 22.80 | 44.67 | 3.48 | 63.19 | 58.91 |
| InternVL-2 | 40B | 80.31 | 89.23 | 82.59 | 50.98 | 85.52 | 76.08 | 85.88 | 98.56 | 79.90 | 99.15 | 62.03 |
| InternVL-1.5 | 26B | 79.49 | 88.90 | 79.00 | 60.70 | 83.70 | 72.50 | 78.90 | 94.50 | 71.40 | 95.80 | 148.10 |
| Qwen2-VL | 7B | 79.14 | 88.17 | 88.17 | 55.68 | 83.32 | 78.86 | 80.73 | 85.57 | 81.20 | 92.13 | 144.36 |
| InternVL-2 | 26B | 78.82 | 88.64 | 82.06 | 48.50 | 84.44 | 72.72 | 83.16 | 97.47 | 77.60 | 110.30 | 80.10 |
| LLaVA-OneVision | 7B | 78.71 | 89.17 | 76.02 | 60.98 | 80.12 | 70.69 | 81.38 | 95.88 | 62.10 | 140.45 | 136.97 |
| InternVL-2 | 8B | 77.84 | 87.90 | 77.00 | 52.02 | 82.48 | 70.65 | 82.25 | 97.03 | 76.50 | 89.77 | 36.70 |
| InternVL-2 | 4B | 73.97 | 87.71 | 74.51 | 38.43 | 81.04 | 65.19 | 78.08 | 94.00 | 75.00 | 54.08 | 30.17 |
| Cambrian | 34B | 73.26 | 88.46 | 72.11 | 52.07 | 74.60 | 51.48 | 80.41 | 85.52 | 59.00 | 8.18 | 6.08 |
| InternVL-2 | 2B | 73.00 | 88.90 | 72.39 | 43.74 | 74.72 | 57.69 | 72.70 | 94.25 | 75.50 | 79.52 | 59.81 |
| Cambrian | 13B | 72.39 | 88.53 | 73.07 | 53.28 | 72.60 | 50.73 | 73.93 | 79.08 | 61.40 | 14.33 | 9.44 |
| XComposer-4KHD | 7B | 71.93 | 87.00 | 74.30 | 51.90 | 80.60 | 72.80 | 34.40 | 96.00 | 66.90 | 134.00 | 111.40 |
| Cambrian | 8B | 71.84 | 88.24 | 72.47 | 52.17 | 73.44 | 48.05 | 72.99 | 80.32 | 61.60 | 9.13 | 7.97 |
| CogVLM-1 | 17B | 71.34 | 88.90 | 79.70 | 46.90 | 67.00 | 63.30 | 61.90 | 70.50 | 59.10 | 28.40 | 44.70 |
| MiniCPM-LLaMA3-2.5 | 8B | 71.22 | 88.00 | 75.00 | 52.30 | 72.90 | 56.90 | 71.70 | 53.00 | 69.40 | 35.50 | 52.90 |
| Qwen2-VL | 2B | 70.86 | 87.78 | 78.70 | 40.59 | 73.12 | 67.15 | 70.21 | 77.89 | 75.30 | 103.52 | 131.92 |
| CogVLM-2 | 19B | 69.93 | 87.56 | 77.59 | 18.51 | 79.84 | 62.62 | 72.41 | 90.93 | 76.60 | 24.10 | 42.23 |
| LLaVA-LLaMA3 | 8B | 69.74 | 87.80 | 65.40 | 60.20 | 69.30 | 37.60 | 71.60 | 73.30 | 55.00 | 135.00 | 69.60 |
| InternVL-2 | 1B | 68.46 | 87.94 | 69.67 | 33.84 | 71.40 | 52.02 | 62.56 | 89.59 | 74.20 | 49.34 | 18.03 |
| LLaVA-1.6-vicuna | 13B | 67.29 | 87.50 | 67.00 | 46.30 | 62.20 | 41.50 | 70.40 | 73.50 | 55.00 | 101.90 | 67.30 |
| LLaVA-1.6 | 34B | 67.22 | 85.60 | 68.90 | 31.00 | 67.40 | 51.90 | 76.10 | 82.70 | 58.60 | 114.40 | 69.10 |
| Mantis-Idefics-2 | 8B | 66.91 | 86.90 | 63.51 | 52.50 | 63.56 | 31.17 | 66.81 | 81.80 | 54.20 | 79.42 | 134.08 |
| Idefics-2 | 8B | 66.73 | 86.80 | 71.30 | 53.90 | 26.40 | 37.00 | 69.20 | 87.20 | 61.60 | 71.90 | 119.10 |
| LLaVA-OneVision | 0.5B | 65.65 | 88.33 | 65.85 | 44.17 | 61.36 | 46.23 | 57.09 | 67.03 | 57.60 | 131.90 | 120.81 |
| Deepseek-VL | 7B | 65.62 | 87.10 | 63.20 | 48.70 | 60.60 | 34.30 | 63.40 | 81.70 | 43.30 | 67.60 | 110.10 |
| LLaVA-1.6-vicuna | 7B | 64.80 | 87.60 | 64.90 | 44.20 | 55.00 | 37.00 | 65.30 | 70.20 | 52.40 | 100.00 | 72.00 |
| LLaVA-1.5 | 13B | 59.58 | 87.10 | 48.70 | 58.30 | 18.10 | 29.50 | 59.40 | 72.80 | 33.60 | 115.40 | 104.00 |
| Mantis-SigLIP | 8B | 58.96 | 81.47 | 49.59 | 52.90 | 42.56 | 26.56 | 57.84 | 75.36 | 34.50 | 91.37 | 111.43 |
| LLaVA-1.5 | 7B | 57.27 | 87.00 | 46.10 | 53.40 | 18.20 | 25.80 | 55.20 | 69.50 | 31.50 | 109.00 | 98.00 |
| MiniCPM | 1.0 | 56.18 | 85.10 | 55.30 | 47.30 | 15.40 | 20.10 | 56.90 | 43.00 | 60.00 | 25.90 | 41.60 |
| InstructBLIP-vicuna | 7B | 47.87 | 85.00 | 33.20 | 45.20 | 12.50 | 22.90 | 34.00 | 36.40 | 25.90 | 141.40 | 74.00 |
| Tiny-LLaVA-1 | 1.4B | 34.30 | 56.30 | 38.50 | 3.80 | 11.10 | 22.20 | 32.30 | 58.20 | 17.20 | 80.90 | 83.10 |

## A.3 Pipeline details

### A.3.1 Prompt template details

**Semi-Automated Screening Process Prompt** We selected GPT-4V as the basis for automatic judgment and interacted with the GPT-4V API using specific prompt templates for different subtasks.

---

**Semi-Automated Screening Process Prompt(VQA tasks)**

Please judge whether the <Answer>is the golden answer to the <Question>. If it is, please reply YES, otherwise reply NO.

<**Question**>: {question} <**Answer**>: {answer}

<**Your judgement**> : <YES or NO>

---

**Semi-Automated Screening Process Prompt(captioning tasks)**

Now there is an image captioning task.
Please first describe the content of the image, then compare the image content with the provided captions.
If the captions are suitable as captions for the image, please answer YES; if they are not suitable, please answer NO.
Respond with NO if any of the captions are unsuitable. Respond with YES only if all captions are suitable.

<**Captions**>: {answer}
<**Description**>: <Content of the image>

<**Your judgement**>: <ONLY YES or NO>

---

**Exact Vision Description Prompt** For the QVD experiment, we use LLaVA-NEXT-110B to extract information from the images, with the following prompt:

---

**Exact Vision Description Prompt**

<**image**> Please provide a description of the following image, You should consider elements in the image.

---

### A.3.2 Metrics

**Subtask metrics**: As shown in the Tab 7, different metrics are used for different subtasks. It is important to note that, except for the CIDEr metric, all other metrics have a range between 0 and 1. The final score for each subtask is calculated by taking the average of these metrics.

Table 7: Metrics for different subtask

| Metric | Subtask | Formula |
|--------|---------|---------|
| **Accuracy** | AI2D, ScienceQA-IMG, OCRBench, POPE | $\text{Accuracy} = \begin{cases} 1, & \text{if the prediction is correct} \\ 0, & \text{if the prediction is incorrect} \end{cases}$ |
| **CIDEr** | TextCaps,COCO-Caption | $\text{CIDEr} = \frac{1}{m}\sum_{i=1}^{m}\sum_{n=1}^{N} w_n \cdot \frac{g_i^{(n)} \cdot r_i^{(n)}}{\|g_i^{(n)}\|\|r_i^{(n)}\|}$ |
| **Match score** | OK-VQA,TextVQA | $\text{SCORE} = \min\left(1, \frac{\text{match\_nums}}{3}\right)$ |
| **ANLS** | InfoVQA | $\text{ANLS}(X,Y) = 1 - \frac{\text{Lev}(X,Y)}{\max(|X|,|Y|)}$ |
| **Relaxed Overall** | ChartQA | $\text{SCORE} = \frac{|\text{prediction}-\text{SCORE}|}{|\text{target}|}$ |

**Overall metric**: For the overall metric, we explored two mainstream calculation methods: arithmetic mean 1 and weighted mean 2.

$$\text{Arithmetic Mean} = \frac{1}{n}\sum_{i=1}^{n} x_i \tag{1}$$

$$\text{Weighted Mean} = \frac{\sum_{i=1}^{n} w_i x_i}{\sum_{i=1}^{n} w_i} \tag{2}$$

The arithmetic mean directly calculates the average of each subtask's scores, while the weighted mean takes into account the number of samples in each subtask. We compare the results of these two calculation methods, as shown in the Tab 8. weighted average method achieves a higher correlation with WV-ELO. This suggests that the weighted average method is slightly superior to the arithmetic mean, as it considers the impact of the number of data points on the overall score, thereby avoiding potential errors caused by uneven data distribution. Therefore, in our work, we ultimately chose the weighted average as the method for calculating the overall score.

Table 8: Comparison of different overall metrics method

| model | overall_weighted | overall_sum | overall_cider | WV_bench |
|---|---|---|---|---|
| LLaVA-1.6-vicuna-7B | 30.15 | 30.46 | 36.07 | 992 |
| LLaVA-1.6-vicuna-13B | 37.08 | 36.52 | 41.04 | 956 |
| LLaVA-1.6-34B | 44.06 | 43.30 | 47.12 | 1059 |
| CogVLM | 51.03 | 47.66 | 44.03 | 1016 |
| Deepseek-VL | 38.1 | 39.04 | 43.31 | 979 |
| Idefics2 | 36.39 | 38.43 | 43.83 | 965 |
| MiniCPM-v-1.0 | 26.15 | 28.95 | 35.79 | 910 |
| Tinny-LLaVA-1-hf | 13.95 | 17.79 | 24.15 | 879 |
| LLaVA-1.5-13B | 20.38 | 22.88 | 32.3 | 891 |
| InstructBLIP-vicuna-7B | 15.55 | 18.61 | 29.56 | 862 |
| correlation score | **0.91** | 0.90 | 0.87 | 1 |

A.3.3 DIFFICULTY CLASSIFICATION DETAILS

For subtasks using the accuracy (acc) metric, where the scores are binary, with only 1 or 0, other tasks may have various possible score distributions (e.g., COCO-Caption, OK-VQA). Therefore, we determine the threshold score based on the overall distribution of subtask scores, and choose the cutoff value that offers the greatest distinction, as shown in Tab 9, for the metrics ANLS, Relaxed Overall and Accuracy (Acc), the threshold is set to 1.0, for BLEU-4 (for the captioning task, we use the BLEU-4 metric to represent the score for each question), the threshold is set to 0.2, while for Match Score, it is set to 0.6. When the score is greater than the threshold, it is marked as correct; otherwise, it is marked as incorrect.

| Metrics | bleu4 | Match score | ANLS | Relaxed Overall | Acc |
|---|---|---|---|---|---|
| **Threshold** | 0.2 | 0.6 | 1.0 | 1.0 | 1.0 |

Table 9: Thresholds for Different Metrics

### A.3.4 RETRIEVE FROM REAL WORLD QUERYD

---

**Qwen2-72B Judge Prompt**

Your task is to compare the content of two questions along with their corresponding image descriptions to determine if they are the same or aligned. Analyze from multiple perspectives, such as theme, question type, and description content.

Please adhere to the following guidelines:

**1. Theme Consistency:**
- Compare whether the themes of the two questions and their corresponding image descriptions match. If they focus on entirely different topics, they should be marked as not aligned.

**2. Question Type:**
- Analyze whether the question types (e.g., technical, artistic, textual) of both questions match with each other and align with their respective image descriptions. If they are of different natures, note the mismatch.

**3. Description Alignment:**
- Compare the task or content expected in each question with what is visually or descriptively present in both image descriptions. If the questions or image content require specific actions (e.g., reading text or coding) that differ from each other or the descriptions, they should be marked as misaligned.

**4. Evaluate Similarity:**
- Rate the similarity between the two questions and their respective descriptions on a scale from 1 to 5, where 1 means entirely different and 5 means highly similar.

**5. Output Clarification:**
- You should return whether the two questions and their image descriptions align or not in a simple "True" or "False" result. - Provide a brief reason for your conclusion. - Include a similarity rating from 1 to 5, based on how well the questions and descriptions match. - The output should only contain the "result," "reason," and "similarity rating" fields.

**### Example:**
**<Question 1>:** Can you write codes to load this 3D object?
**<Description 1>:** The image shows a stone sculpture of an angel sitting on a pedestal. The angel has large, feathered wings that spread out behind it, and its head is bowed down, as if in deep thought or prayer. The angel's body is draped in flowing robes, and its arms are crossed over its lap. The pedestal is ornately carved with intricate designs, and the entire sculpture is set against a dark background, which makes the white stone stand out even more. The overall mood of the image is one of solemnity and reverence.

**<Question 2>:** What is written in the image?
**<Description 2>:** The image shows the word "ART" in white capital letters on a blue background. The letters are bold and have a slight shadow effect, giving them a three-dimensional appearance. The overall design is simple and modern, with a focus on the text itself.

**Result:** False
**Reason:** The first question asks for coding assistance to load a 3D object, but its description is about an angel sculpture. The second question is focused on reading text from an image, which is aligned with its description showing the word "ART." The themes, questions, and descriptions are entirely different.
**Similarity Rating:** 1

**<Input Question 1>:** {Question 1}
**<Input Description 1>:** {Description 1}

**<Input Question 2>:** {Question 2}
**<Input Description 2>:** {Description 2}

**<Output>:**

---

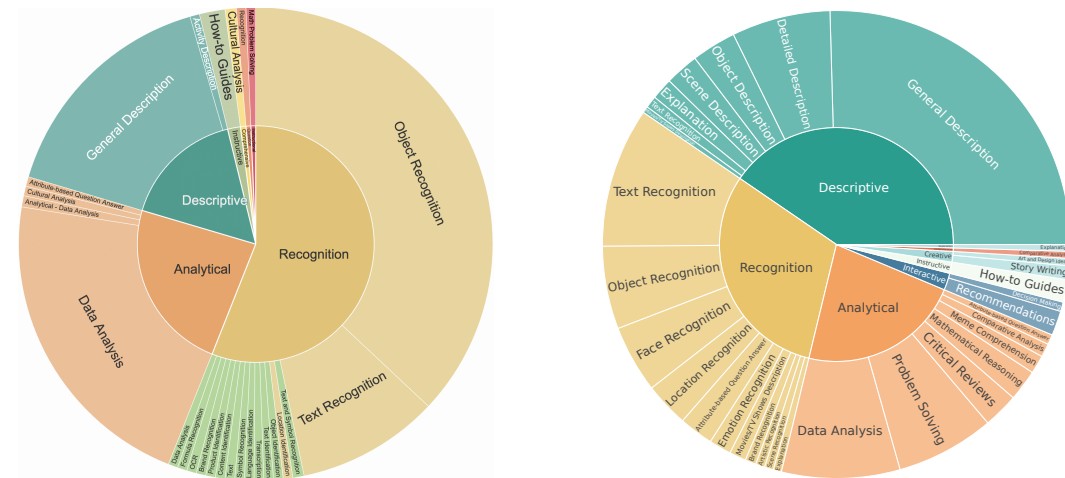

Figure 9: category difference between LIME-fit and wildvision bench

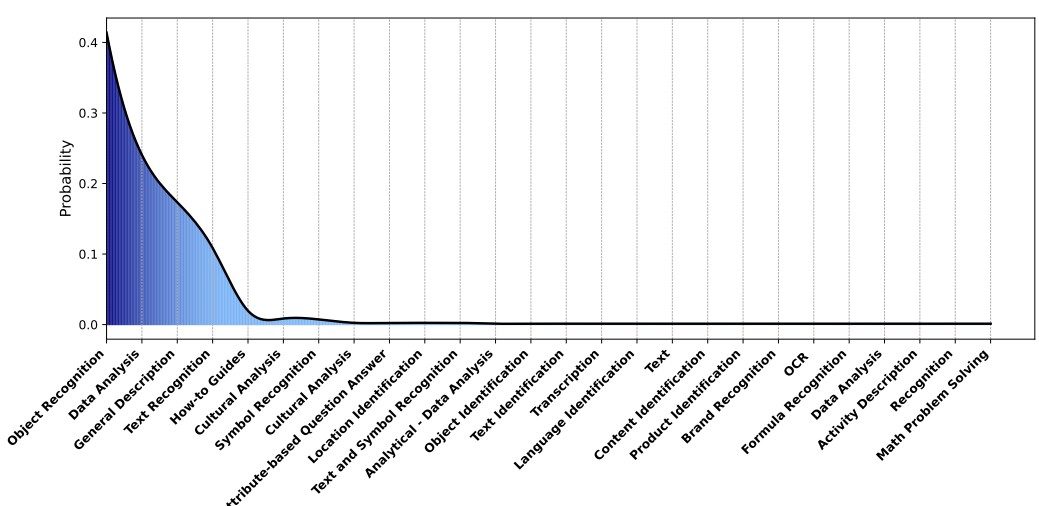

Figure 10: subcategory distrubution of LIME-fit.

## A.4 ABLATION STUDY ABOUT DATA SIZE

Table 10: data size ablation study on OK-VQA.

| Model | Full | 100 | 500 | 1200 |
|---|---|---|---|---|
| llava1.5-7B | 22.71 | 17.00 | 20.76 | 22.92 |
| llava1.5-13B | 31.59 | 36.00 | 29.60 | 30.23 |
| llava1.6-7B | 11.46 | 13.00 | 10.40 | 11.32 |
| llava-llama3-8B | 36.12 | 32.60 | 36.92 | 36.17 |
| xcomposer2-4khd | 25.91 | 29.40 | 26.48 | 26.90 |
| minicpm | 25.76 | 20.60 | 29.92 | 25.30 |
| instructblip | 21.45 | 20.60 | 23.60 | 21.78 |
| idefics2 | 32.76 | 27.60 | 35.12 | 33.00 |
| internvl | 38.36 | 45.00 | 39.80 | 38.28 |

Table 11: data size ablation study on ChartQA.

| Model | Full | 100 | 500 | 1200 |
|---|---|---|---|---|
| llava1.5-7B | 4.77 | 3.00 | 3.80 | 4.17 |
| llava1.5-13B | 4.71 | 5.00 | 4.40 | 4.33 |
| llava1.6-7B | 42.81 | 39.00 | 42.00 | 42.67 |
| llava-llama3-8B | 64.78 | 66.00 | 66.00 | 65.75 |
| xcomposer2-4khd | 82.11 | 80.00 | 83.00 | 82.92 |
| minicpm | 70.37 | 67.00 | 71.60 | 70.75 |
| instructblip | 2.95 | 3.00 | 3.00 | 3.00 |
| idefics2 | 13.18 | 16.00 | 12.80 | 14.25 |
| internvl | 87.13 | 89.00 | 87.80 | 86.92 |

Table 12: data size ablation study on TextVQA.

| Model | Full | 100 | 500 | 1200 |
|---|---|---|---|---|
| llava1.5-7B | 16.68 | 14.40 | 18.34 | 17.46 |
| llava1.5-13B | 19.54 | 17.90 | 22.30 | 20.14 |
| llava1.6-7B | 38.58 | 43.00 | 38.62 | 39.35 |
| llava-llama3-8B | 39.81 | 46.40 | 38.26 | 40.17 |
| xcomposer2-4khd | 61.20 | 59.40 | 60.98 | 61.63 |
| minicpm | 63.07 | 60.30 | 63.90 | 63.39 |
| instructblip | 11.66 | 8.60 | 12.00 | 11.25 |
| idefics2 | 55.94 | 54.90 | 57.76 | 56.56 |
| internvl | 70.28 | 70.10 | 70.48 | 70.74 |

Table 13: data size ablation study on InfoVQA.

| Model | Full | 100 | 500 | 1200 |
|---|---|---|---|---|
| llava1.5-7B | 9.40 | 7.00 | 9.00 | 8.83 |
| llava1.5-13B | 12.18 | 16.00 | 11.60 | 11.17 |
| llava1.6-7B | 21.30 | 19.00 | 23.00 | 20.33 |
| llava-llama3-8B | 22.69 | 25.00 | 23.40 | 22.33 |
| xcomposer2-4khd | 72.36 | 72.00 | 75.80 | 73.75 |
| minicpm | 49.22 | 55.00 | 48.20 | 48.75 |
| instructblip | 9.73 | 11.00 | 8.40 | 9.50 |
| idefics2 | 24.69 | 23.00 | 24.00 | 25.42 |
| internvl | 72.08 | 69.00 | 72.20 | 72.25 |

# B  CASE STUDY

The original dataset contains noise data. In the following figure, we categorize the problematic data into three types and present specific examples from different datasets.

**Text Answerable Questions:**  Some questions can be answered without the need for visual information, mainly focusing on the AI2D and ScienceQA datasets. As shown in figs. 30 and 31, AI2D and ScienceQA emphasize knowledge in the field of science while overlooking the importance of visual information. Given the background of domain knowledge, some LLMs are able to provide answers even without requiring visual input.

**Annotation Error Questions:**  Most benchmarks are manually curated, which inevitably leads to annotation errors. Problematic questions exist in almost all benchmarks. It can be found in figs. 32, 33 and 39 to 44.

**Repeated Question:**  Some benchmarks also contain a significant amount of duplicate data, where the question content and image content are completely identical. This issue is mainly found in the POPE dataset, as shown in the figs. 34 to 38.

**List of Case Study Figures**

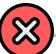 **MMMU**

**Question:** What vessel(s) serve(s) areas involved in speech in the majority of people? <image 1>

**Options:** ['Right middle cerebral artery.', 'Left middle cerebral artery.', 'Right and left middle cerebral arteries.', 'Right and left posterior cerebral arteries.']

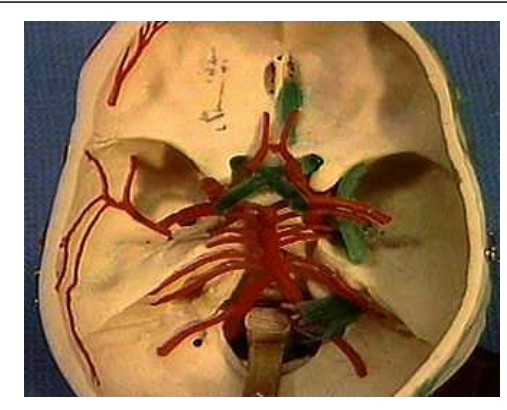

**Error Category:** Answer Leakage

**Ground Truth:** Left middle cerebral artery.

Figure 11: A sample bad case of MMMU
Back to List of figures

**MMMU**

**Question:** Which of the following does the offspring of a pod bug resemble?

**Options:** ['Similar to the adult, but shorter and without wings', 'Grub', 'Maggot', 'Caterpillar', "Don't know and don't want to guess"]

Error Category: Answer Leakage

Ground Truth: Similar to the adult, but shorter and without wings

Figure 12: A sample bad case of MMMU
Back to List of figures

**MMMU**

**Question:** 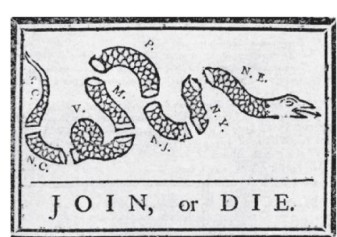 <image 2> Which of the following Acts of Parliament was passed in direct response to the events of the Boston Tea Party?

**Options:** ['Coercive Acts', 'Tea Act', 'Townshend Acts', 'Currency Act']

*Join, or Die*, Benjamin Franklin, 1754

"It is in vain, sir, to extenuate the matter. Gentlemen may cry, 'Peace, Peace,' but there is no peace. The war is actually begun! The next gale that sweeps from the north will bring to our ears the clash of resounding arms! Our brethren are already in the field! Why stand we here idle? What is it that gentlemen wish? What would they have? Is life so dear, or peace so sweet, as to be purchased at the price of chains and slavery? Forbid it, Almighty God! I know not what course others may take; but as for me, give me liberty or give me death."

Patrick Henry, speaking at the Second Virginia Convention, 1775

<image 1>

<image 2>

**Error Category: Answer Leakage**

**Ground Truth: Coercive Acts**

Figure 13: A sample bad case of MMMU
Back to List of figures

## MMMU

**Question:** Which theory of <image 1> focuses on the labels acquired through the educational process?

**Options:** ['Critical sociology', 'Feminist theory', 'Functionalist theory', 'Symbolic interactionism']

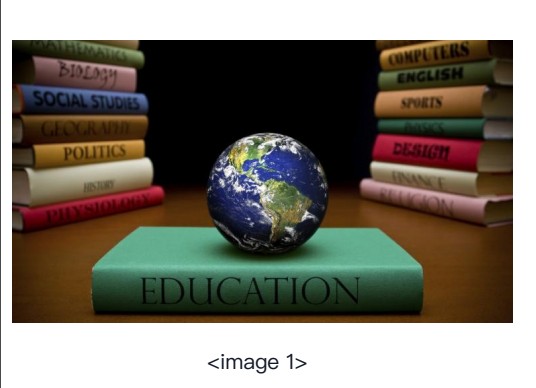

<image 1>

**Error Category: Answer Leakage**

**Ground Truth: Symbolic interactionism**

Figure 14: A sample bad case of MMMU

## MMMU

**Question:** Hicks Products produces and sells patio furniture through a national dealership network. They purchase raw materials from a variety of suppliers and all manufacturing, and assembly work is performed at their plant outside of Cleveland, Ohio. They recorded these costs for the year ending December 31, 2017. What is total revenue?

**Options:** [A:'$3,100,000', B:'$2,616,000', C:'$2,474,000', D:'$484,000']

| | |
|---|---|
| Sales revenue | $3,100,000 |
| Straight-line depreciation on office equipment | 90,000 |
| Advertising and marketing expense | 625,000 |
| Administrative salaries | 136,000 |
| Cost of goods sold | 1,700,000 |
| Rent on corporate headquarters | 65,000 |

< 11 >

**Error Category: Easy Question**

**Ground Truth:  A**

Figure 15: A easy sample of MMMU
Back to List of figures

## MMMU

**Question:** You are asked to compare two options with parameters as given. The risk-free interest rate should be assumed to be 6%. Assume the stocks on which these options are written pay no dividends. <image 1> Which call option is written on the stock with the higher volatility?

**Options:** [A:'A', B:'B', C:'Not enough information']

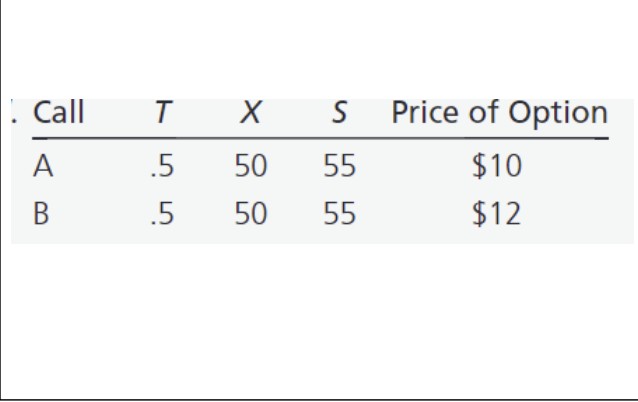

| . Call | T | X | S | Price of Option |
|--------|-----|----|----|-----------------|
| A | .5 | 50 | 55 | $10 |
| B | .5 | 50 | 55 | $12 |

< 28 >

**Error Category: Easy Question**

**Ground Truth: B**

Figure 16: A easy sample of MMMU
Back to List of figures

## MMMU

**Question:** <image 1> What seems to be the issue with this young citrus tree?

**Options:** [A:'Mineral deficiency', B:'Nematode attack', C:"Don't know and don't want to guess", D:'There is no problem', E:'Pot bound']

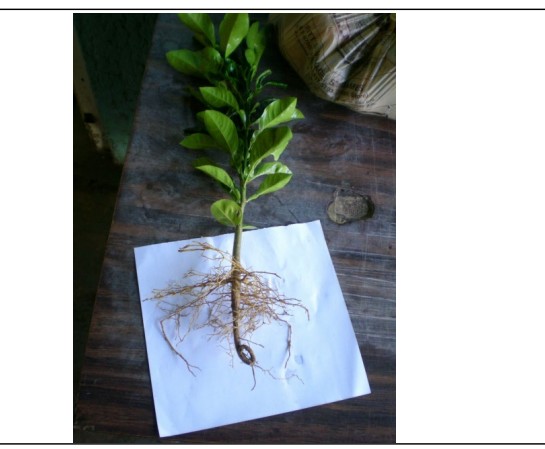

< 33 >

**Error Category: Easy Question**

**Ground Truth: E**

Figure 17: A easy sample of MMMU
Back to List of figures

**MMMU**

**Question:** <image 1> What is the common term for the yellow area surrounding the site of an infection?

**Options:** [A:'I don't know and I don't want to guess', B:'Corona', C:'Border', D:'Halo', E:'Toxin zone']

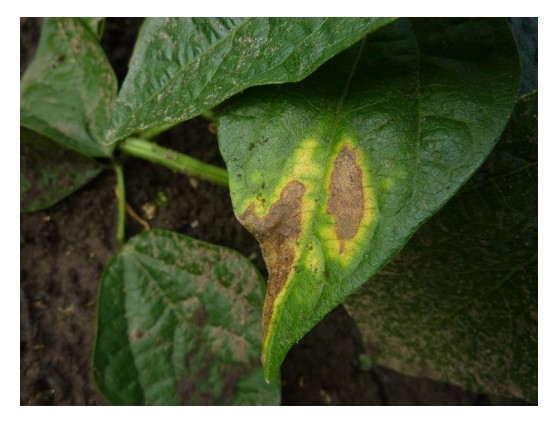

< 45 >

**Error Category: Easy Question**

**Ground Truth: D**

Figure 18: A easy sample of MMMU
Back to List of figures

**MMMU**

**Question:** <image 1> What is the substance present on the top surface of these citrus leaves?

**Options:** [A:'Algae', B:"Don't know and I don't want to guess", C:'Honey dew', 'Gummosis-produced resin', 'Bacterial ooze']

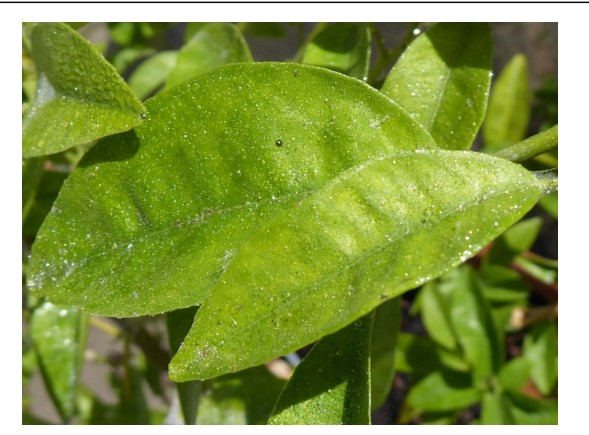

< 47 >

**Error Category: Easy Question**

**Ground Truth: C**

Figure 19: A easy sample of MMMU
Back to List of figures

**MMBench**

**Question:** Complete the statement. Ammonia is ().

**Options:** [A:'an elementary substance', B:'a compound']

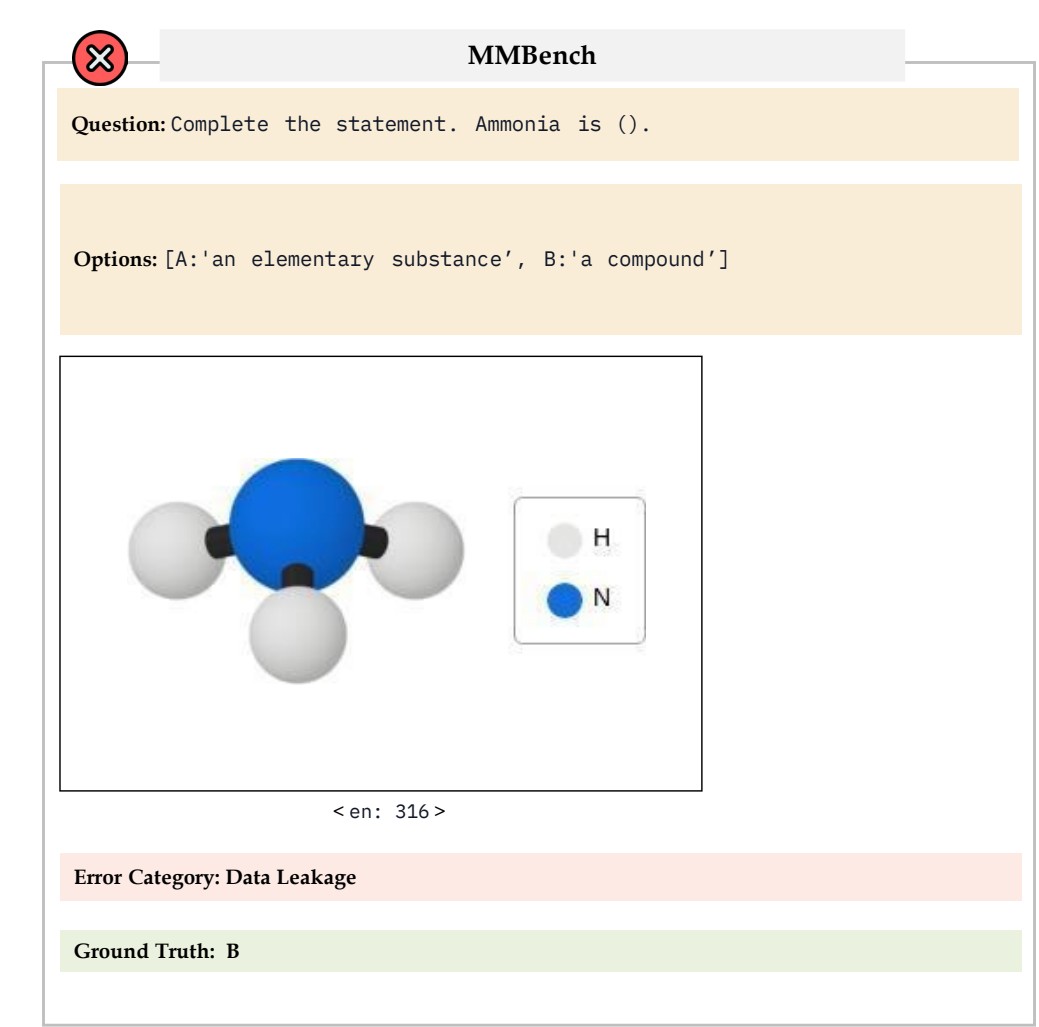

< en: 316 >

Error Category: Data Leakage

Ground Truth: B

Figure 20: A sample bad case of MMBench

## MMBench

**Question:** Identify the question that Madelyn and Tucker's experiment can best answer.

**Options:** [A:'Does Madelyn's snowboard slide down a hill in less time when it has a thin layer of wax or a thick layer of wax?', B:' Does Madelyn's snowboard slide down a hill in less time when it has a layer of wax or when it does not have a layer of wax?']

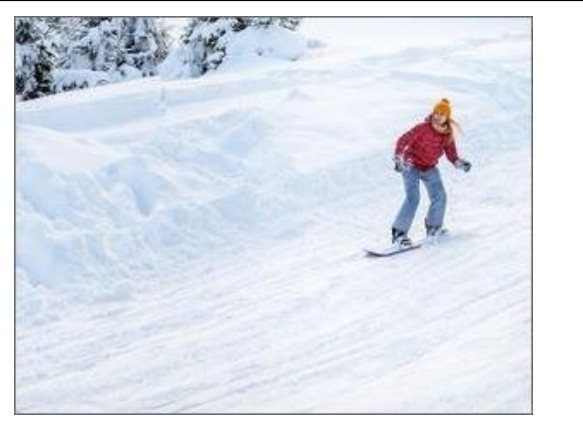

< en: 241 >

**Error Category: Data Leakage**

**Ground Truth: B**

Figure 21: A sample bad case of MMBench

Figure 22: A sample bad case of MMBench
Back to List of figures

## MMBench

**Question:** Which animal's skin is also adapted for survival in cold places?

**Options:** [A:'fantastic leaf-tailed gecko', B:'polar bear']

< en: 278 >

**Error Category: Data Leakage**

**Ground Truth: B**

Figure 23: A sample bad case of MMBench
Back to List of figures

## MMBench

**Question:** Which material is this spatula made of?

**Options:** [A:'rubber', B:'cotton']

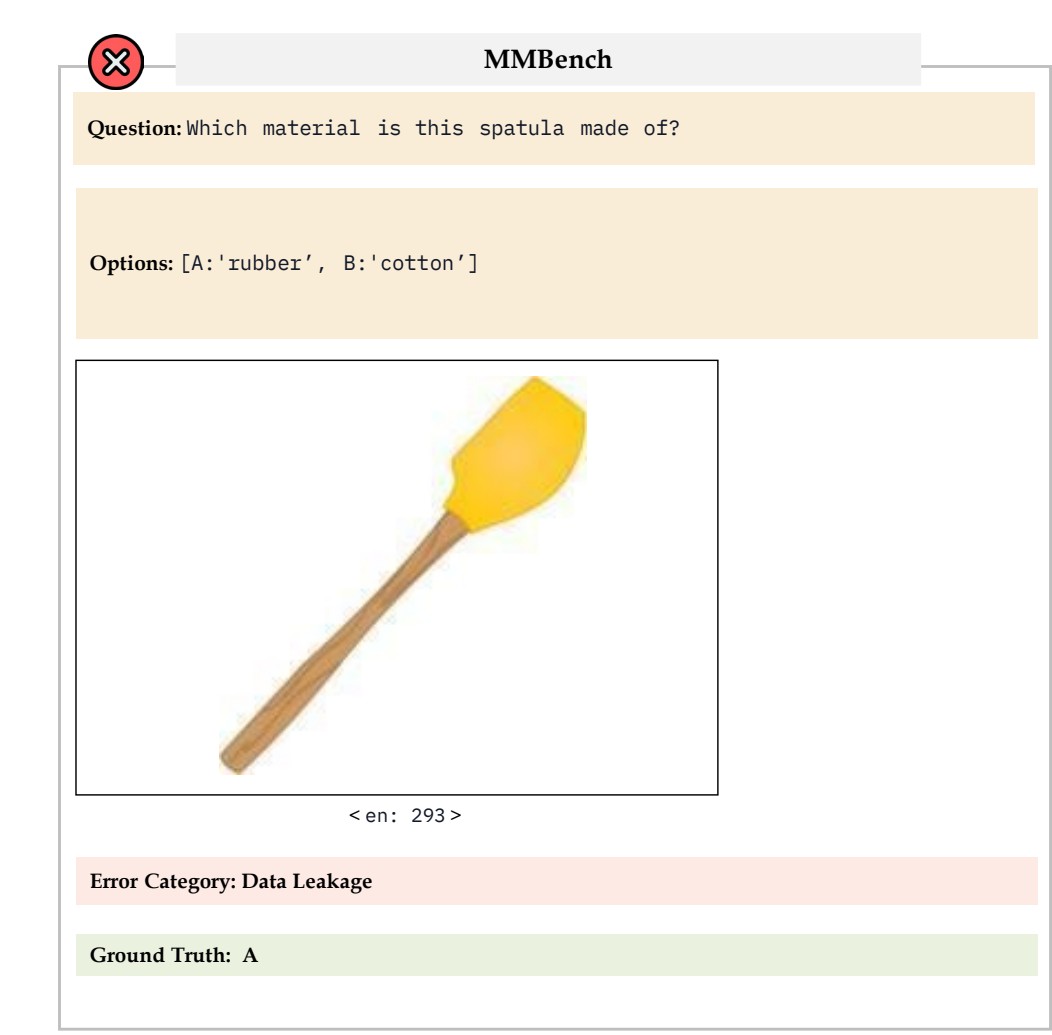

< en: 293 >

Error Category: Data Leakage

Ground Truth:  A

Figure 24: A sample bad case of MMBench
Back to List of figures

## MMBench

**Question:** 图中所示建筑名称为?

**Options:** [A:天坛， B:故宫， C:黄鹤楼， D:少林寺]

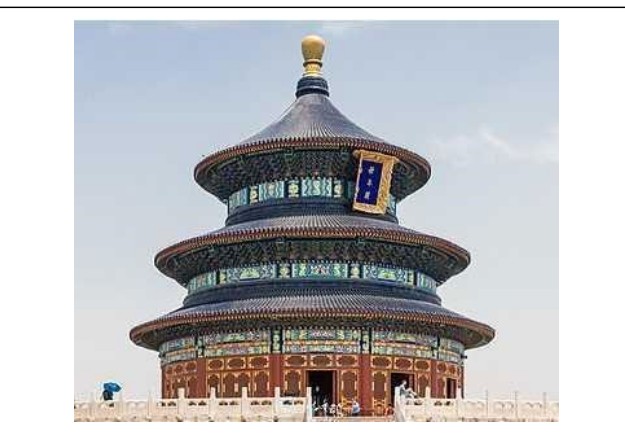

< CC: 0 >

**Error Category: Easy Question**

**Ground Truth: A**

Figure 25: A easy sample of MMBench
Back to List of figures

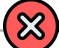 **MMBench**

**Question:** 图中所示建筑名称为？

**Options:** [A:东方明珠，B:长城，C:中山陵，D:少林寺]

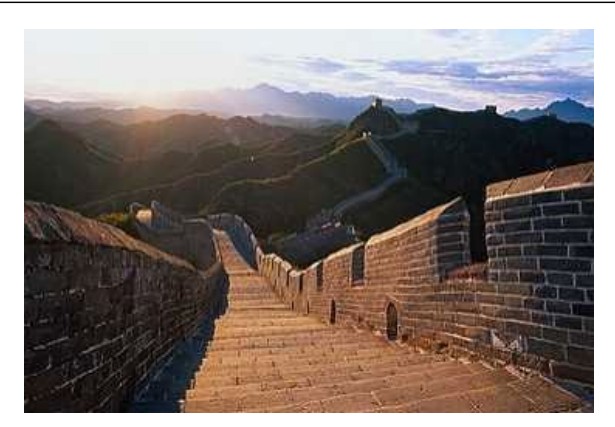

< cc: 1 >

**Error Category: Easy Question**

**Ground Truth: B**

Figure 26: A easy sample of MMBench
Back to List of figures

## MMBench

**Question:** 图中所示景观所在地点为？

**Options:** [A:重庆，B:香港，C:青岛，D:上海]

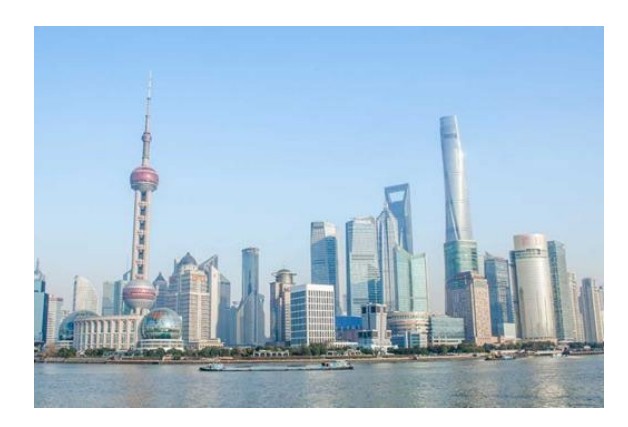

< cc: 4 >

**Error Category: Easy Question**

**Ground Truth: D**

Figure 27: A easy sample of MMBench
Back to List of figures

## ❌ MMBench

**Question:** Which of the following could Laura and Isabella's test show?

**Options:** [A:'if the concrete from each batch took the same amount of time to dry', B:'if a new batch of concrete was firm enough to use']

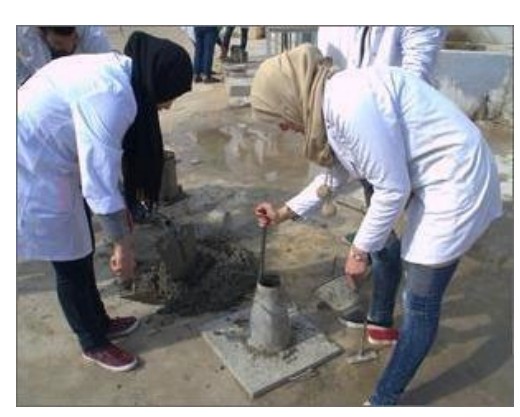

< cc: 1 >

**Error Category: Easy Question**

**Ground Truth: B**

Figure 28: A easy sample of MMBench
Back to List of figures

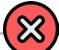

## MMBench

**Question:** Which animal's limbs are also adapted for gliding?

**Options:** [A:"northern flying squirrel', B: ring-tailed lemur']

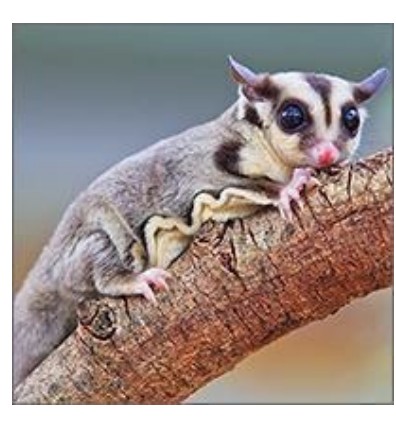

< cc: 9 >

**Error Category: Easy Question**

**Ground Truth: A**

Figure 29: A easy sample of MMBench
Back to List of figures

**AI2D**

**Question:** Which stage follows the egg stage of development in a beetle's life cycle?

**Options:** ["Nymph", "Larva", "Adule", "Pupa"]

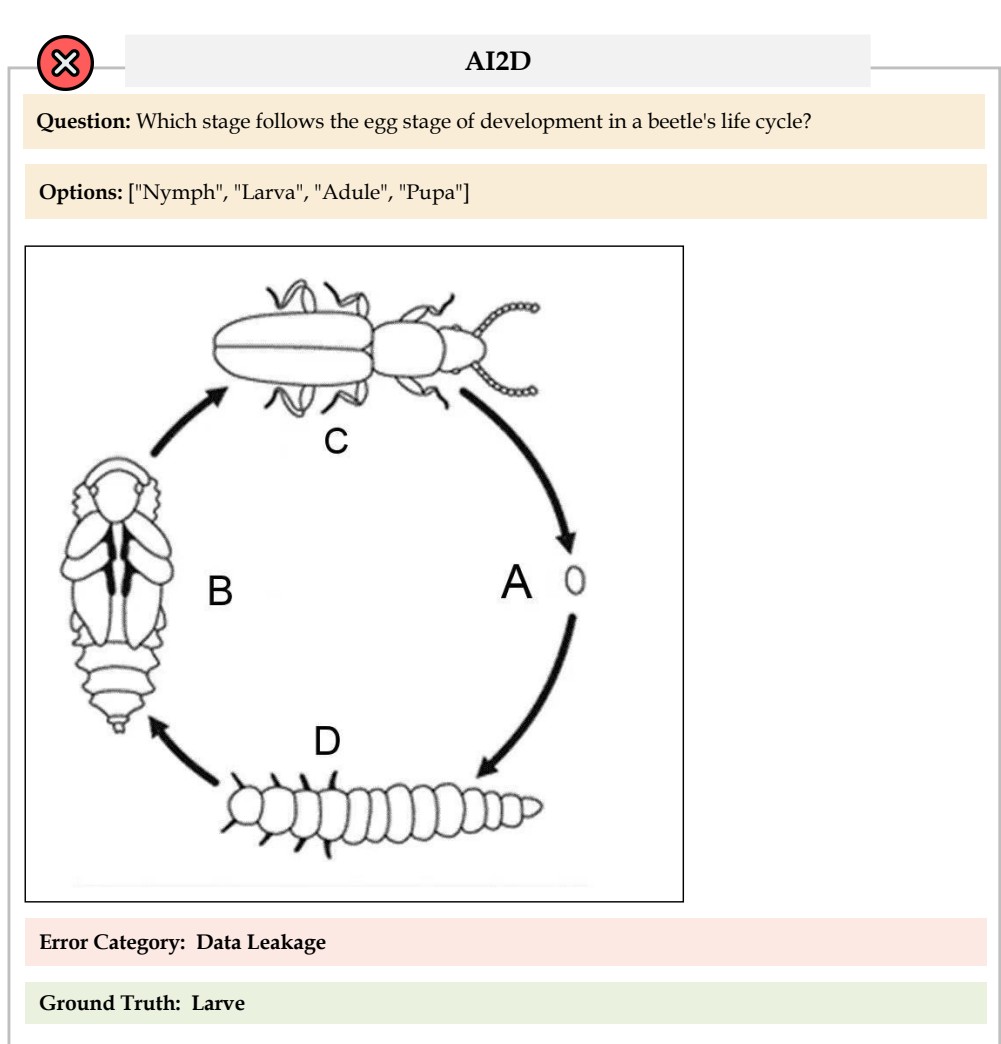

**Error Category:  Data Leakage**

**Ground Truth:  Larve**

Figure 30: A sample bad case of AI2D

**AI2D**

**Question:** In the illustration, if mahi mahi were to die off the large shark population would?

**Options:** [ "decrease", "remain the same", "can't tell", "increase" ]

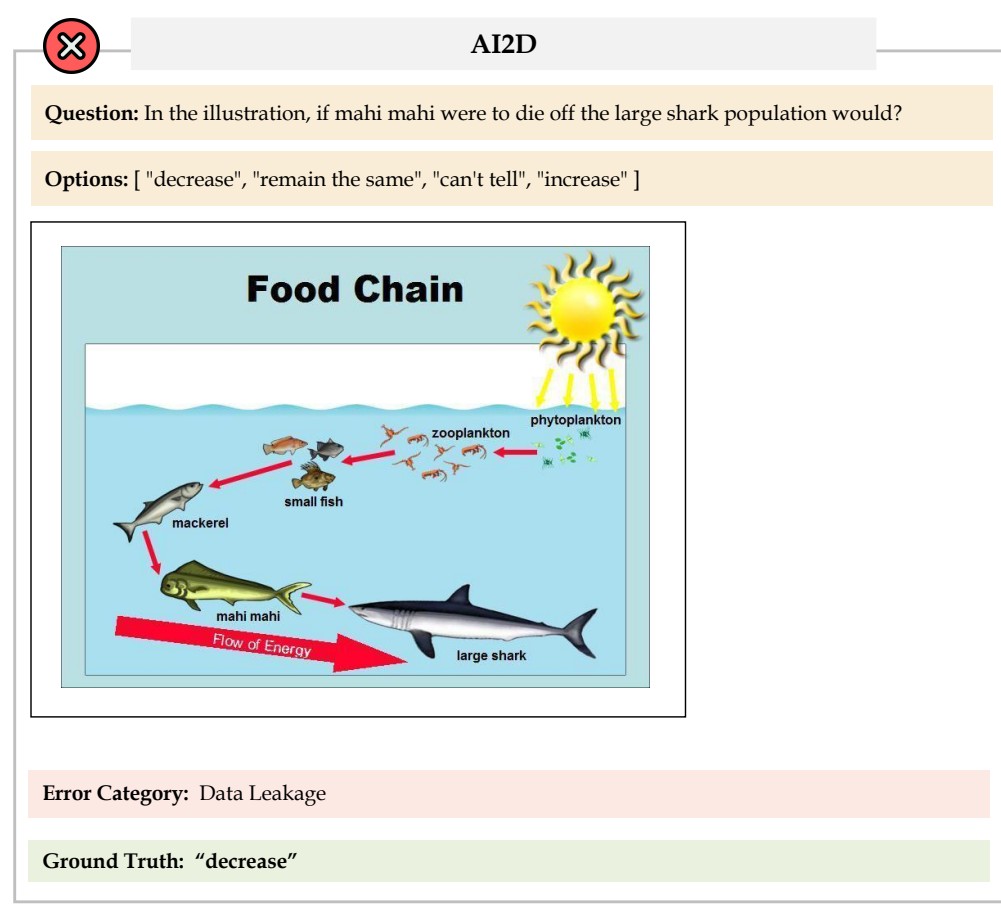

**Error Category:** Data Leakage

**Ground Truth:** "decrease"

Figure 31: A sample bad case of AI2D

**InfographicVQA**

**Question:** What percent of executives does not use social media daily?

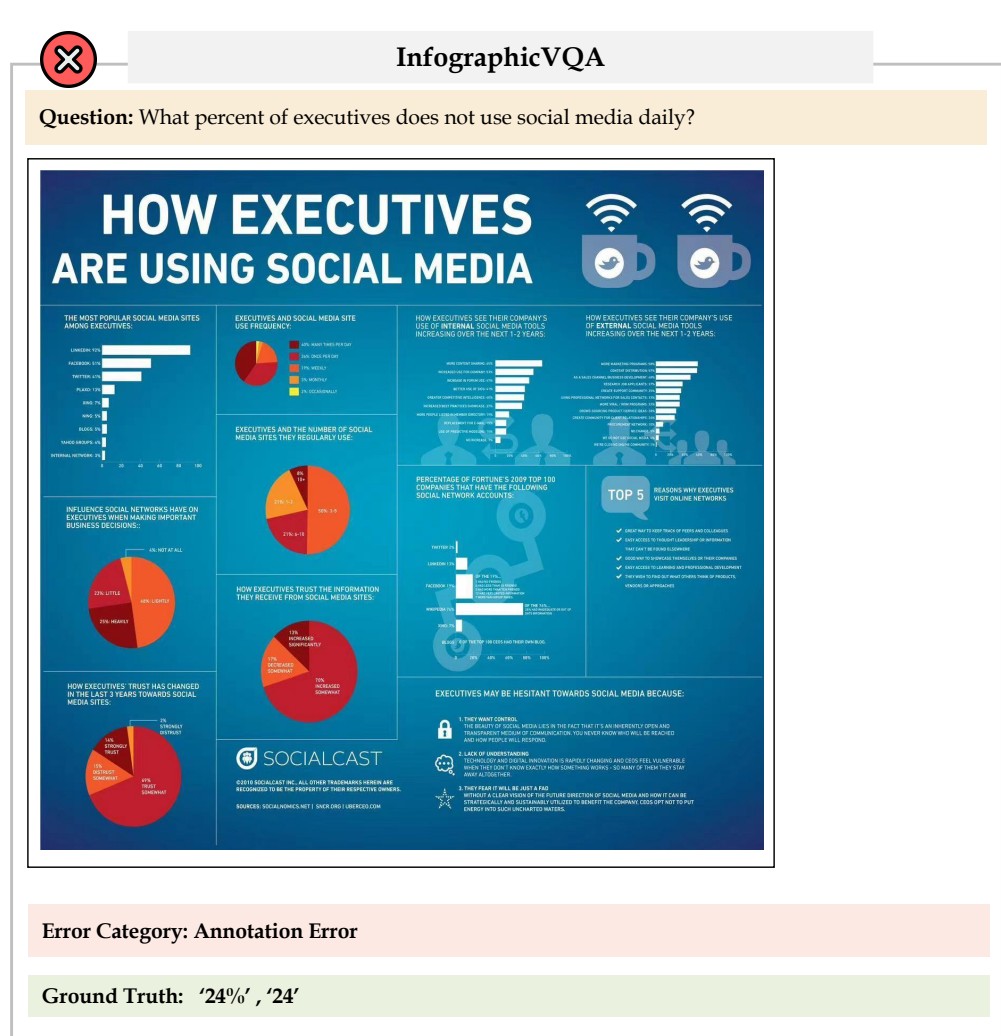

**Error Category: Annotation Error**

**Ground Truth:** '24%' , '24'

Figure 32: A sample bad case of InfoVQA

**InfographicVQA**

**Question:** What is the second last solution given?

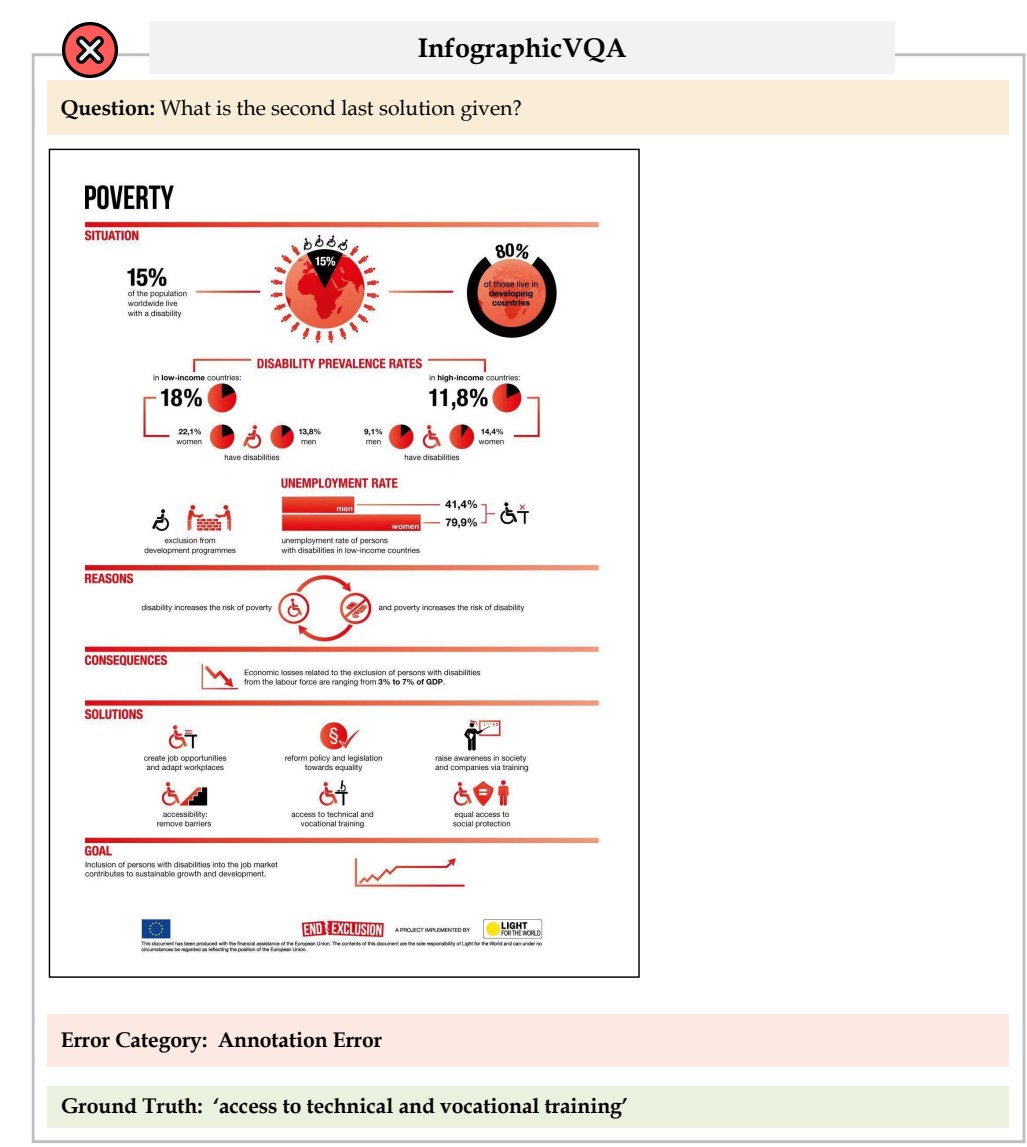

**Error Category: Annotation Error**

**Ground Truth: 'access to technical and vocational training'**

Figure 33: A sample bad case of InfoVQA

**POPE**

**Question::** Is there a tv in the image?

**Options:** Yes

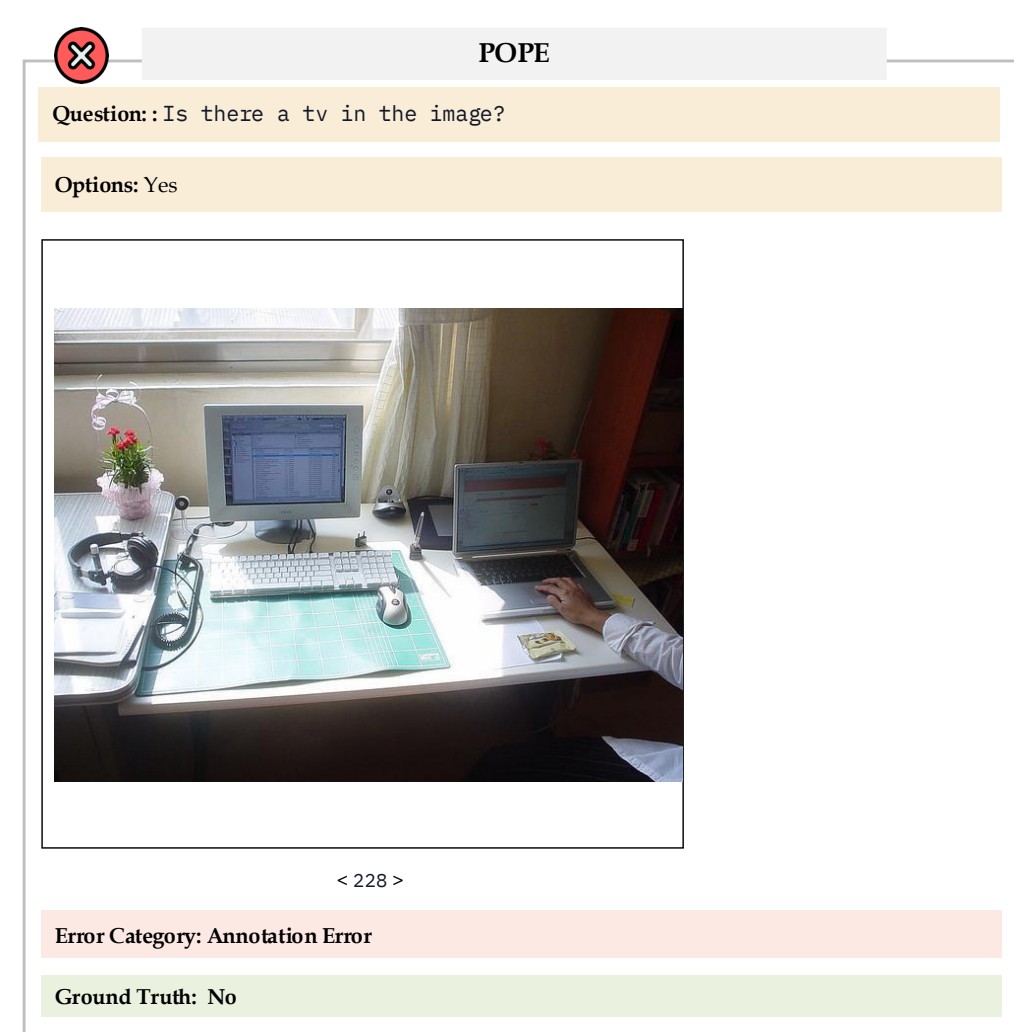

< 228 >

**Error Category: Annotation Error**

**Ground Truth: No**

Figure 34: A sample bad case of POPE
Back to List of figures

**POPE**

**Question::** Is there a dining table in the image?

**Options:** Yes

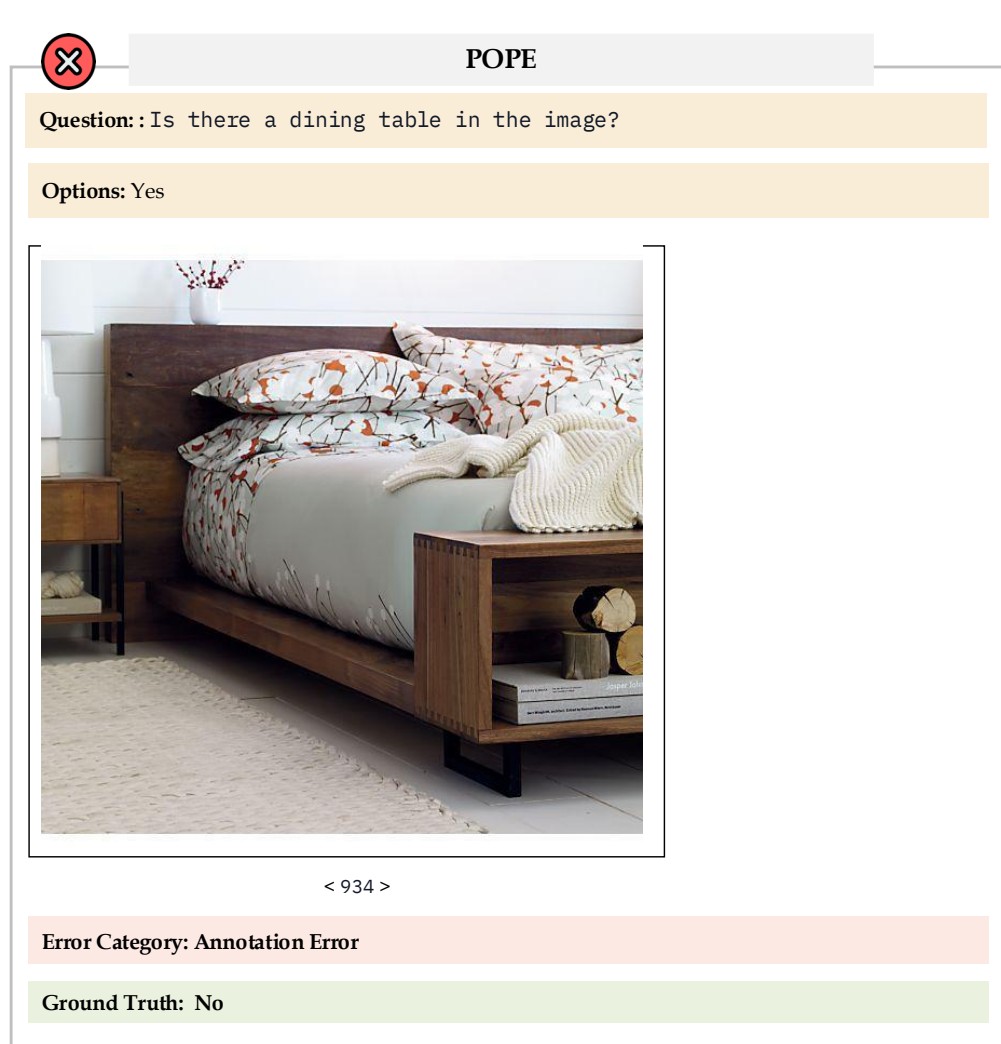

< 934 >

**Error Category: Annotation Error**

**Ground Truth:  No**

Figure 35: A sample bad case of POPE
Back to List of figures

**POPE**

**Question::** Is there a boat in the image?

**Options:** Yes

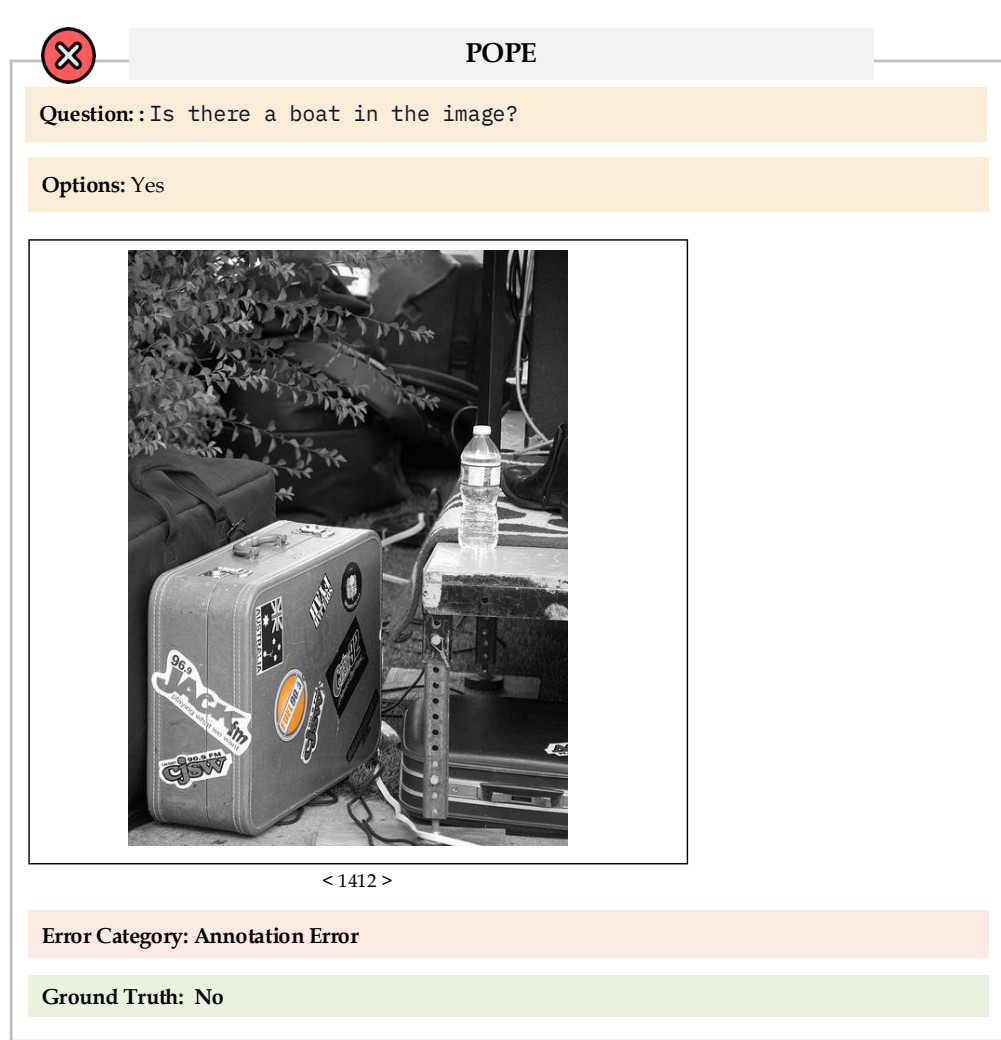

< 1412 >

**Error Category:** Annotation Error

**Ground Truth:** No

Figure 36: A sample bad case of POPE

**POPE**

**Question::** Is there a boat in the image?

**Options:** Yes

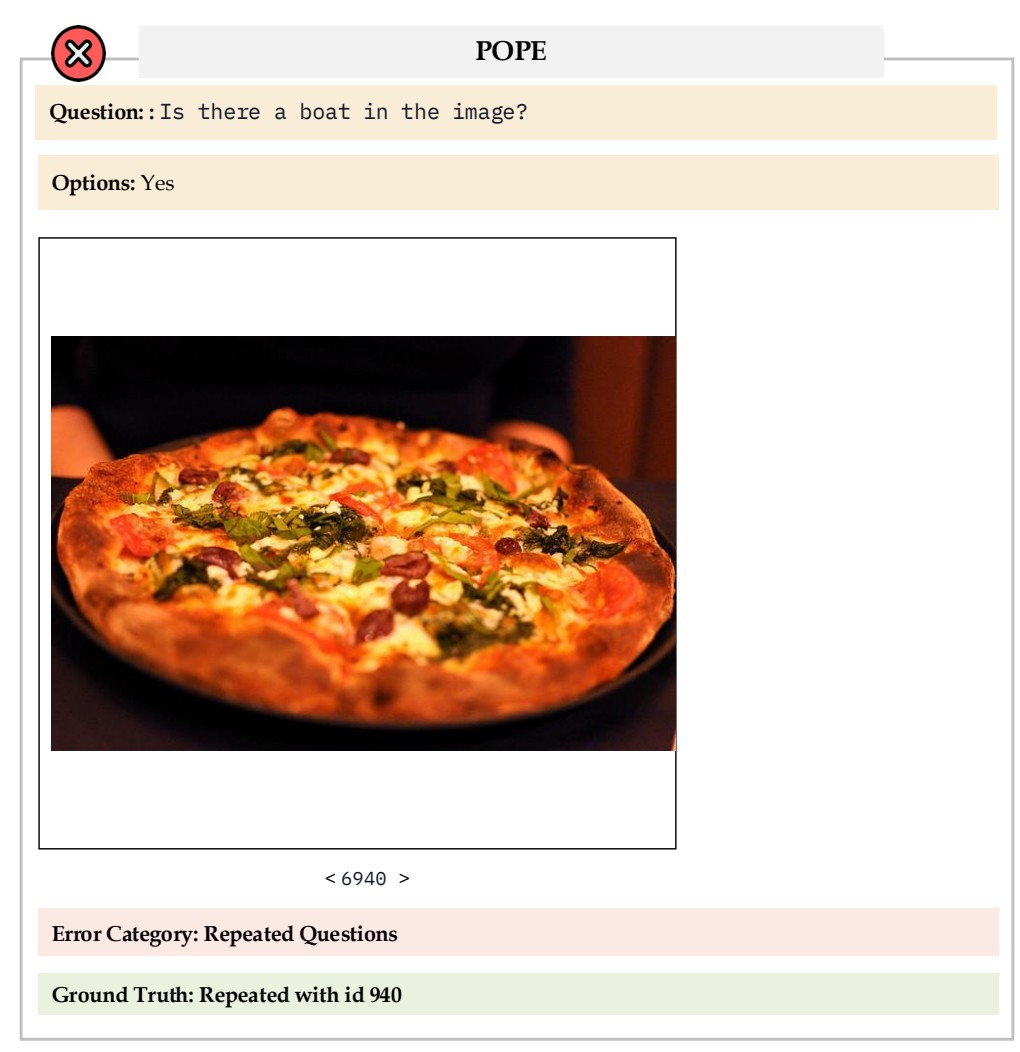

< 6940 >

**Error Category: Repeated Questions**

**Ground Truth: Repeated with id 940**

Figure 37: A sample bad case of POPE
Back to List of figures

**POPE**

**Question::** Is there a dining table in the image?

**Options:** Yes

< 6694 >

**Error Category: Repeated Questions**

**Ground Truth: Repeated with id 694**

Figure 38: A sample bad case of POPE
Back to List of figures

## OK VQA

**Question::** How would you dress for this setting?

**Options:** [ "shorts", "shorts", "shorts", "shorts", "bathing suit", "bathing suit", "bikini", "bikini", "summer", "summer" ]

< 1708495 >

**Error Category:** Annotation Error

**Ground Truth:** [ "shorts", "swimming suit", "bathing suit", "bikini" ]

Figure 39: A sample bad case of OKVQA
Back to List of figures

## OK VQA

**Question::** Where are these people?

**Options:** [ "outside", "outside", "outside", "outside", "field", "field", "on hill", "on hill", "outdoors", "outdoors" ]

< 3981385 >

**Error Category:** Annotation Error

**Ground Truth:** [ "outside", "riverbank", "grassland", "field", "hill", "outdoors", "lawn" ]

Figure 40: A sample bad case of OKVQA
Back to List of figures

## OK VQA

**Question::** How is this effect painted on to walls?

**Options:** [ "sponge", "sponge", "sponge", "sponge", "with sponge", "with sponge", "sponged", "sponged", "sky", "sky" ]

< 1269585 >

**Error Category:** Annotation Error

**Ground Truth:** [ "whitewash", "paint", "plaster" ]

Figure 41: A sample bad case of OKVQA
Back to List of figures

## Text VQA

**Question::** what is one of the numbers on the buttons of the calculator?

**Options:** [ "1", "1", "1", "1", "1", "7", "7", "5", "1", "5" ]

< 35925 >

**Error Category:** Annotation Error

**Ground Truth:** [ "1", "2", "3", "4", "5", "6", "7", "8", "9", "0" ]

Figure 42: A sample bad case of TextVQA
Back to List of figures

**Text VQA**

**Question::** what is served at this place?

**Options:** [ "gift certificates", "ice cream, coffee, and sandwiches", "ice cream& coffee", "traditional italian ice cream and coffee", "ice cream & coffee", "ice cream, coffee, and grilled focaccia sandwiches", "ice cream & coffee", "traditional italian, ice cream and coffee, grilled focaccia sandwiches", "ice cream & coffee, grilled focaccia sandwiches", "gelato" ]

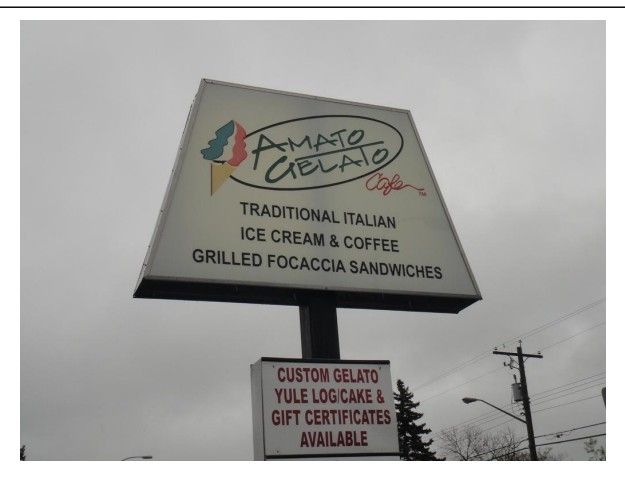

< 37706 >

**Error Category:** Annotation Error

**Ground Truth:** [ "ice cream", "coffee", "sandwiches", "gelato", "cake", "yule log", "gift certificates" , "grilled focaccia sandwiches"]

Figure 43: A sample bad case of TextVQA
Back to List of figures

**Text VQA**

**Question::** what is the cell phone carrier?

**Options:** [ "cingular", "blackberry", "cingular", "cingular", "cingular", "cingular", "at&t", "cingular", "cingular", "cingular" ]

< 36711 >

**Error Category:** Annotation Error

**Ground Truth:** [ "EDGE " ]

Figure 44: A sample bad case of TextVQA
Back to List of figures