# OpenReview forum: "LIME: LESS IS MORE FOR MLLM EVALUATION"
_ICLR.cc/2025/Conference — Submitted to ICLR 2025_

### Official Review · Reviewer_Mt3s · 2024-10-17

**Soundness:** 3
**Presentation:** 3
**Contribution:** 3
**Rating:** 6
**Confidence:** 5

**Summary:**

This paper presents a refined and efficient MLLM benchmark called LIME, which enhances the quality of existing benchmarks through semi-automatic refinement. LIME consists of 9,400 evaluation samples across six types of tasks and ten different benchmark datasets. The authors evaluated over 30 models and provided some analyses based on the results.

**Strengths:**

1. The removal of easy samples and meaningless or erroneous data from the dataset is crucial for more efficient and reasonable evaluation of MLLMs. The authors utilize GPT-4V and human annotators to fillter out those illogical and meaningless questions, which seems to have been overlooked in previous benchmarks.
2. The authors evaluate over 30 baseline models and provide an analysis of MLLM performance based on the evaluation results, which clearly represents a significant amount of work.
3. The authors also construct a similarity search system for investigating the gap between LIME and real-world users’ queries, which shows that the current benchmark does not fully cover the instruction requirements of real-world scenarios.

**Weaknesses:**

1. The proposed benchmark, LME, integrates existing benchmarks and adopts their evaluation metrics, which have been previously criticized in earlier works (specifically designed for evaluating MLLMs) [1, 2] as being unsuitable for assessing open-form outputs of MLLMs. For instance, the authors mention, “for tasks such as AI2D, ScienceQA, OCRBench, and POPE, we calculate the accuracy of the extracted responses.” In these benchmarks, if the correct answer is "bike" but the model outputs "bicycle," it is considered incorrect, which is an unreasonable approach. The authors should employ more appropriate evaluation metrics, such as multiple-choice questions, true/false questions, or scoring by GPT.
2. To eliminate answer leakage, such as when a model has seen the questions during training, the authors conduct a text-only check using pure text LLMs. Based on the responses from these LLMs, they remove samples that can be directly answered without using the image. However, this approach is unreasonable because these multimodal questions would only appear in the training of MLLMs. Therefore, the authors should use MLLMs to filter out such questions instead of relying on LLMs.

[1] SEED-Bench: Benchmarking Multimodal LLMs with Generative Comprehension
[2] MMBench: Is Your Multi-modal Model an All-around Player?

**Questions:**

Please see weaknesses.

---

> ### Author Response · Authors · 2024-11-25
> **Response to Reviewer Mt3s**
>
> Thank you very much for your valuable feedback, it is highly beneficial for further improving our work. Below, we will provide a detailed response to the issues you have raised.
> # Response to Question 1
>
> As shown in Table 1 of our paper, most of the selected tasks are objective questions such as multiple-choice questions or T/F reasoning tasks. In these tasks (AI2D, ScienceQA, POPE), there is no need to account for variations in sentence structure or wording that convey similar meanings to the golden answer (e.g., "bike" and "bicycle" being equivalent). We require MLLMs to provide answers in a predefined format (e.g., selecting A, B, C, or D for multiple-choice questions).  and as for the OCR task, the model is expected to extract text directly from the image, meaning the generated text must match the image exactly; otherwise, any variation, even if semantically similar, is incorrect. Therefore, using accuracy (acc) to evaluate these four tasks is both reasonable and accurate.
>
> For most evaluation tasks in LIME, the answers are typically precise and unique, so GPT-eval is not essential, and it also comes with significant costs. The goal of LIME is to make the evaluation of MLLMs faster and more efficient, which is why we have retained the direct calculation of accuracy.
>
>
> # Response to Question 2
>
> Thank you very much for your suggestions regarding mitigating data leakage in MLLMs.  In fact, regarding "**these multimodal questions would only appear in the training of MLLMs**," we have eliminated these quesions through the Semi-Automated Screening Process. In the [General Response](https://openreview.net/forum?id=3c4zQpIFNK&noteId=vceOK3COLP), we have provided more detailed explanations and experiments. As shown in the Table, even the SOTA MLLMs (Qwen2-VL-7B  and InternVL-2-8B ) achieve extremely low scores in the text-only setting. This demonstrates that our pipeline is robust enough to address all potential data leakage, ensuring there is no more data leakage in LIME.
>
> Apologies for the unclear expression in our previous submission, which may have caused a misunderstanding for you. We have refined this explanation to clarify our approach to eliminating answer leakage and to address any potential concerns.
>
> If you have any further questions, please feel free to let us know. We remain open and eager to address any further concerns or questions you may have.

---

> > ### Comment · Reviewer_Mt3s · 2024-11-26
> >
> > The authors' response addressed my concerns. I decided to improve my rating.

---

> > > ### Author Response · Authors · 2024-11-26
> > > **Thank You For Your Reponse**
> > >
> > > Thank you sincerely for your positive and constructive feedback on our submission.

---

### Official Review · Reviewer_bgCw · 2024-10-31

**Soundness:** 3
**Presentation:** 3
**Contribution:** 3
**Rating:** 8
**Confidence:** 4

**Summary:**

The paper proposes the LIME, a refined and efficient benchmark for MLLM evaluation. The paper first shows that existing benchmarks contain a large proportion of easy or noise samples that cannot reflect the actual capabilities of MLLM. Then, the paper proposes a three-stage pipeline to filter the existing 10 benchmarks across 6 types. The easy samples, wrong-labeled samples, and answer-leakage samples are removed during this process. The refined benchmark can provide a more rigorous evaluation of the existing MLLMs.

**Strengths:**

1. This paper uncovers the problem of existing benchmarks and the proposed filter method is reasonable and meaningful.
2. The filter benchmark provides a more rigorous evaluation of the existing MLLMs and will have practical significance for future MLLM evaluations.
3. The experiment results are comprehensive and insightful.

**Weaknesses:**

1. Do not compare with other general MLLM benchmarks like MMMU or MMBench. I would also like to see whether the easy samples or answer-leakage samples exist in these benchmarks.

**Questions:**

1. In Line 036, the author mentions 'How many chairs in the image'. Does it mean all existing MLLMs' counting capabilities are not satisfactory?
2. In Line 156, 'The model has encountered a specific question during training'. Does the term 'model' here refer to LLM or MLLMs?

---

> ### Author Response · Authors · 2024-11-25
> **Response to Reviewer bgCw**
>
> # Response to Weakness 1:
>
> Thank you very much for your valuable feedback. Below, we present a comparison of the similarities and differences between LIME and MMMU/MMBench and provide detailed case studies in the appendix B.
>
> **MMMU**[1] is a large-scale, multidisciplinary, multimodal understanding and reasoning benchmark, which contains 11,500 samples across six different categories. It focuses on evaluating the logic and reasoning capabilities of MLLMs. However, some studies have pointed out that MMMU heavily relies on pure text-based knowledge and suffers from potential data leakage issues[3,4], failing to adequately assess the multimodal capabilities of MLLMs. In contrast, LIME employs a rigorous data processing pipeline to eliminate potential data leakage issues, providing more challenging tasks for multimodal models.
>
> **MMBench**[2] is designed to evaluate MLLMs across six different capability dimensions, MMBench leverages LLMs and MLLMs as judges to filter question sources. However, it only removes text-only and incorrect samples, lacking fine-grained difficulty categorization and still posing risks of potential data leakage, as a result, MMBench is unable to effectively distinguish the performance differences between different models. As shown in below Table, compared to MMBench,  LIME can better reflect the capability differences among various models.
>
> We have posted the cases of easy samples and answer leakage in MMMU and MMBench in Appendix B.
>
>
> | Model              | MMMU (val) | MM_bench (test_en) | LIME (overall) |
> |-|-|-|-|
> | InternVL-2-2B       | 36.3      | 73.4            | 53.64          |
> | InternVL-2-8B       | 51.2      | 82      | 62             |
> | LLaVA-1.6-vicuna-7B        | 69.2   | 67.09             | 30.15          |
> | LLaVA-1.6-vicuna-13B       | 70    | 69.15             | 37.08          |
> | Qwen2-VL-2B        | 42.2        | 74.6              | 54             |
> | Qwen2-VL-7B         | 53.7    | 82.8         | 65.28          |
>
> *Specifically, to ensure fairness, we recorded the MMMU and MMBench scores of different models from the OpenVLM Leaderboard.*
>
> [1] MMMU: A Massive Multi-discipline Multimodal Understanding and Reasoning Benchmark for Expert AGI
>
> [2] MMBench: Is Your Multi-modal Model an All-around Player?
>
> [3] MMMU-Pro: A More Robust Multi-discipline Multimodal Understanding Benchmark
>
> [4] Cambrian-1: A Fully Open, Vision-Centric Exploration of Multimodal LLMs
>
> # Response to Question 1:
>
> We point out "**How many chairs are in the image**"  to highlight that such inquiries focus only on the superficial aspects of the image, such as counting and object recognition. They lack a deeper understanding and reasoning about the image's content, which poses a more significant challenge for MLLMs. We hope our response could address your concerns.
>
> # Response to Question 2:
>
>
> The term 'model' here specifically refers to **MLLMs**, and we primarily focus on the potential risk of data leakage during the training process of MLLMs. Thank you very much for your suggestions regarding the content of the paper, and we have refined this expression in the paper to make it clearer, explicitly denoting MLLMs.

---

### Official Review · Reviewer_1M8b · 2024-11-01

**Soundness:** 3
**Presentation:** 3
**Contribution:** 3
**Rating:** 5
**Confidence:** 3

**Summary:**

Existing MLLM benchmarks often include overly simple or uninformative samples, making it difficult to effectively distinguish the performance of different MLLMs. This work proposes LIME , a refined and efficient benchmark curated using a semi-automated pipeline.
This pipeline filters out uninformative samples and eliminates answer leakage by focusing on tasks that require image-based understanding.  The experiments show that LIME reduces the number of samples by 76% and evaluation time by 77%, while it seems promising to be more effective for distinguishing different models’ abilities.

**Strengths:**

The problem is important and interesting to the community. Evaluation is an important part for multimodal LLM. This work dives deep into existing benchmarks and conducts comprehensive analysis to study the specific questions in those benchmarks. The motivation of Figure 1 and 2 is clear and important.

**Weaknesses:**

1. My biggest concern is that the approach only filter the samples from the existing benchmarks, do we need to consider adding other metrics/domains to evaluate MLLMs?
2. Another interesting thing is that sometimes MLLM may not "read" image but directly answer the questions based on the knowledge from LLM, do we need to consider adding this into the benchmark?

**Questions:**

See weakness.

---

> ### Author Response · Authors · 2024-11-25
> **Response to Reviewer 1M8b**
>
> Thank you so much for your valuable suggestions, which have provided significant inspiration and direction for our research.Below, we offer a detailed explanation to address your concerns
> # Response to Question 1:
> Our main purpose is to propose a data process pipeline to compress various benchmarks (filtering relatively hard and simple samples, mitigating answer leakage) and curate the LIME to better distinguish MLLMs’ ability with less computation cost. Creating new benchmark data for other domains is important, but that is a separate and independent idea, not the main focus of our work.
>
> Furthermore, this pipeline is designed with plug-and-play architecture, which means that it can be directly applied to other benchmarks of other domains. As for the extending data, we can also use this pipeline to extract more effective samples for evaluating MLLMs.
>
>
> # Response to Question 2:
> Actually, we consider the ‘**MLLM may not "read" image but directly answer the questions based on the knowledge from LLM**’.
> we test the text-only performance of sota MLLMs on LIME. Please refer to the result in  [General Response](https://openreview.net/forum?id=3c4zQpIFNK&noteId=vceOK3COLP), even the SOTA MLLMs(Qwen2-VL-7B ,InternVL-2-8B  ) achieve extremely low scores in the text-only setting, which indicates that our pipeline is robust enough to account for all potential data leakage, and there is no more data leakage in LIME.

---

> > ### Comment · Reviewer_1M8b · 2024-11-26
> > **Thank you for your clarification.**
> >
> > Thank you for your response. I have read all comments and feedback from authors and other reviewers as well. I agree with Reviewer Pra9. We have too many benchmarks every day but this work is not novel for the "core" MLLM evaluation tasks. Therefore, I keep the score with Reviewer Pra9.
> >
> > >Given the rapid developments in this field, several recent publications have explored similar methodologies. I'm wondering if the authors could further elaborate on what distinguishes their approach and its specific contributions to advancing MLLM research. Benchmarks are coming out almost every day, whether designed for new capabilities or a careful mixture of the existing ones.

---

> ### Author Response · Authors · 2024-11-29
> **Response to Reviewer 1M8b**
>
> Thank you very much for your thorough evaluation and feedback on our work. We have provided detailed responses in the [General Response](https://openreview.net/forum?id=3c4zQpIFNK&noteId=lnhOyv6dSU).
>
> 1. about "core" MLLM evaluation tasks
> > “We have too many benchmarks every day but this work is not novel for the "core" MLLM evaluation tasks.”
>
> **The core purpose of LIME is not to propose new MLLM evaluation tasks, but rather to offer a universal pipeline & guideline.** It is a well-established fact that new benchmarks emerge every day. However, most benchmark creation processes encounter similar challenges. **LIME offers a robust pipeline designed to eliminate potential errors in the data and uncover the most 'core' aspects of a benchmark.** As new benchmarks continue to emerge in the future, we are committed to updating LIME every few months (LIME-v2, LIME-v3, etc.), providing increasingly valuable test sets."
>
> We hope our response addresses your concerns. If you have any further questions, we would be very happy to engage in further discussions.

---

> ### Author Response · Authors · 2024-12-02
> **Kindly Reminder to Reviewer 1M8b**
>
> Thank you again for taking your valuable time to review our paper, and we have responded in detail to the concerns you have raised. As the deadline approaches, we kindly request your feedback on our rebuttal. We are eager to have further discussions and address any additional questions you may have.

---

### Official Review · Reviewer_Pra9 · 2024-11-01

**Soundness:** 3
**Presentation:** 3
**Contribution:** 2
**Rating:** 5
**Confidence:** 4

**Summary:**

This paper introduces LIME (Less Is More for MLLM Evaluation), a refined benchmark for evaluating Multimodal Large Language Models (MLLMs). The authors propose a semi-automated pipeline to curate a more efficient and discriminative evaluation dataset by filtering out uninformative samples and eliminating answer leakage. The resulting benchmark reduces the number of evaluation samples by 76% and evaluation time by 77% while maintaining or improving the ability to distinguish between different models' capabilities. Key findings include the inadequacy of traditional automatic metrics for captioning tasks and the importance of excluding caption task scores for more accurate overall performance assessment.

**Strengths:**

* Originality:
   * Novel approach to benchmark curation that focuses on quality over quantity.
   * Creative use of MLLMs themselves as judges for data filtering.
   * Innovative three-stage filtering pipeline (model judgment, semi-automated screening, leakage elimination)
* Clarity:
   * Well-structured presentation of the methodology
   * Clear visualization of data statistics and filtering results
* Quality:
   * Comprehensive empirical validation across multiple models and benchmarks
   * Thorough analysis of the correlation between different subtasks

**Weaknesses:**

- The filtering pipeline heavily relies on existing MLLMs as judges, which could potentially introduce biases from these models into the benchmark. While the authors attempt to mitigate this by using multiple models, a more thorough analysis of potential inherited biases would strengthen the work.
- The paper does not fully explore whether the reduced dataset size might affect the statistical significance of evaluation results. While efficiency gains are clear, more discussion of the tradeoffs between dataset size and evaluation reliability would be valuable
- The choice of tasks and task weightings in the final benchmark appears somewhat arbitrary. A more systematic approach to determining which tasks are most important for evaluating MLLMs would strengthen the methodology.

**Questions:**

1. How sensitive is the filtering pipeline to the choice of judge models? Would using different combinations of models as judges result in significantly different benchmark compositions?
2. How do you ensure that the filtering process doesn't inadvertently favor certain types of model architectures or training approaches?
3. Have you explored whether the reduced dataset size affects the statistical significance of model comparisons? What is the minimum number of samples needed for reliable evaluation?
4. (Minor) If the benchmark is accepted, what will the authors do to let the community buy the idea using your combined filtered benchmark instead of the existing ones? While I believe the benchmark is useful. One concern from my side is that people may still stick to the individual raw datasets.

---

> ### Author Response · Authors · 2024-11-25
> **Response to Reviewer Pra9 (Part 1/n)**
>
> Thank you for your thoughtful and detailed feedback regarding the choice of judge models. This is indeed a highly valuable and thought-provoking question. Below, we provide more detailed experimental evidence to explore the potential impacts of different judge model selections.
> # Response to Question 1 and Question 2.
> The primary goal of the filtering process is to identify samples that most MLLMs can either successfully answer or fail to answer, ensuring the benchmark aligns with the general capabilities and biases of these models. To achieve this, we employ **9 mainstream** MLLMs, including LLaVA, Internvl, MiniCPM, and XComposer, as judge models, using a voting mechanism to filter the samples. This approach mitigates the influence of any single model's bias—such as one model excelling at ScienceQA—on the final results, as the biases of different models tend to balance each other out.
>
> To further explore the influence of different judge model selections, we select **9 completely different** models (Internvl2.Cambrian,Deepseek-VL,CogVLM,... ) and re-perform the filtering process. We then compare the filtering results with those from LIME, We use **Jaccard similarity** and **repetition rate** to measure the overlap between the two distributions
>
> The Jaccard similarity between two sets $A$ and $B$ is defined as:
>
> $$ J(A, B) = \frac{|A \cap B|}{|A \cup B|} $$
>
> The results show that for all subtasks, the Jaccard correlation before and after filtering is **greater than 50%**, indicating that even when selecting entirely different model combinations, the final benchmark set still exhibits high relevance.
>
> | Benchmark         | Jaccard Similarity | Repetition Rate |
> |-|-|-|
> | InfoVQA           | 72.80%             | 95.27%          |
> | AI2D              | 65.14%             | 90.21%          |
> | OCRBench          | 62.98%             | 92.83%          |
> | POPE              | 60.17%             | 78.10%          |
> | ScienceQA     | 58.66%             | 83.66%          |
> | ChartQA           | 54.20%             | 64.41%          |
> | OK-VQA    | 52.67%             | 97.41%          |
> | TextVQA       | 53.37%             | 79.20%          |

---

> ### Author Response · Authors · 2024-11-25
> **Response to Reviewer Pra9 (Part 2/n)**
>
> # Response to Question 3
>
> That's a great question, actually, we determine our dataset size by comparing the performance of 9 MLLMs on datasets of different sizes and evaluating their scores across these data sizes. We calculate the average score gap between each dataset size and the full size (the un-sampled dataset).
>
> As shown in below Table , when the dataset size is 1200, compared to other dataset sizes, the performance gap between the subtask-specific subset and the full dataset decreases to approximately 0.5 points, which is considered acceptable. We also compute the correlations between the subsets of different subtasks and the full dataset: ChartQA: 99.99, TextVQA: 99.99, InfoVQA: 99.97, OK-VQA: 99.7, the results show a strong correlation between the sampled subsets and the full dataset.
>
> | Sub Task    | 100       | 500       | 1200      |
> |-|-|-|-|
> | ChartQA     | 1.92 | 0.73 | 0.51 |
> | InfoVQA     | 2.56 | 1.11 | 0.65 |
> | OK-VQA      | 4.056 | 1.83 | 0.43  |
> | TextVQA     | 2.64  | 1.05 | 0.53 |
>
>
> Considering the trade-off between evaluation resources and evaluation accuracy, as a result,we finally select the 1.2k dataset size, which uses fewer evaluation resources while maintaining evaluation accuracy. the full ablation results can be found in Appendix A.4.
>
> # Response to Question 4
>
> It depends on the situation, studies that just focus on a specific task (e.g., image caption) may not need to assess the model’s performance on our LIME benchmark. In contrast, for fundamental MLLMs, running the model on our benchmark would be convenient to comprehensively evaluate its performance.
>
> Furthermore, with the improvement of MLLMs' capabilities, the original benchmark can no longer reflect the differences in performance between model, benchmarks such as POPE and ScienceQA contain a large number of easy samples, LIME could better distinguish the model’s ability on different domains, which is more helpful for researchers to refine the model’s ability.

---

> ### Author Response · Authors · 2024-11-25
> **Response to Reviewer Pra9 (Part 3/n)**
>
> # Response to Ethics Review.
> Thank you very much for your attention to and suggestions regarding the Ethics Review. We sincerely apologize for previously overlooking this aspect. Below, we have listed the licenses for all benchmarks covered by LIME, and we have also cited all original papers for these datasets, ensuring proper acknowledgment of the authors' contributions.
>
> Licenses of our used dataset are among (Attribution-Sharealike 4.0 International), (MIT License), (Berkeley Software Distribution) and (Apache License Version 2.0), which allow reusers to distribute, remix, adapt, and build upon the material in any medium or format, so long as attribution is given to the creator.
>
> | Dataset           | License                                 |
> |-|-|
> | OCRBench          | Attribution-Sharealike 4.0 International |
> | POPE              | MIT License                             |
> | TextCaps     | Attribution-Sharealike 4.0 International |
> | COCO-Caption     | Berkeley Software Distribution          |
> | TextVQA          | Attribution-Sharealike 4.0 International |
> | OK-VQA            | Apache License Version 2.0             |
> | ChartQA          | Attribution-Sharealike 4.0 International |
> | InfoVQA          | Attribution-Sharealike 4.0 International |
> | ScienceQA         | MIT License                             |
> | AI2D              | Attribution-Sharealike 4.0 International |
>
> Within the constraints of limited time and resources, we have made our best effort to address and explain the concerns you raised. We hope our response alleviates your worries.

---

> > ### Comment · Reviewer_Pra9 · 2024-11-26
> > **Thank you for the rebuttal**
> >
> > Thank you for addressing part of my concerns through the clarification about the pipeline. I appreciate how this approach contributes to data filtering for MLLM evaluation, and I can see its practical utility. While I maintain my original score, I want to acknowledge this positive aspect of the work.
> >
> > Given the rapid developments in this field, several recent publications have explored similar methodologies. I'm wondering if the authors could further elaborate on what distinguishes their approach and its specific contributions to advancing MLLM research. Benchmarks are coming out almost every day, whether designed for new capabilities or a careful mixture of the existing ones.
> >
> > When considering the overall paper alongside other reviewers' comments, I believe there may be opportunities to strengthen the research questions to better highlight the work's novel contributions to the field. A systematic justification, as well as a clearer motivation, are needed. Once someday, the community will realize there is another benchmark needed for the "core" MLLM evaluation tasks, then the proposed one will become less meaningful.

---

> > > ### Author Response · Authors · 2024-11-29
> > > **Thank you for the suggestions**
> > >
> > > Thank you very much for your positive recognition of our work. Your constructive feedback is of great importance to us, as it not only helps improve our current work but also contributes significantly to the development of the MLLM evaluation field. In the [General Response](https://openreview.net/forum?id=3c4zQpIFNK&noteId=lnhOyv6dSU), we have provided detailed replies to the questions you raised, and we hope our answers will address your concerns. Should you have any further comments or questions, we would be more than happy to continue the discussion.

---

> ### Author Response · Authors · 2024-12-02
> **Kindly Reminder to Reviewer Pra9**
>
> Thank you again for taking your valuable time to review our paper, and we have responded in detail to the concerns you have raised. As the deadline approaches, we kindly request your feedback on our rebuttal. We are eager to have further discussions and address any additional questions you may have.

---

### Public Comment · ~Lai_Wei7 · 2024-11-16
**Can we also use the proposed semi-automated pipeline in LLM domain?**

Thanks for your interesting work! LIME is very useful. I wonder whether we can also directly use the proposed semi-automated pipeline in LLM domain. Is LIME specifically designed for MLLMs?

---

> ### Author Response · Authors · 2024-11-25
> **Response to Public Comment**
>
> Thanks for your interest in our work. The answer is definitely yes!  The semi-automated pipeline of LIME is designed with plug-and-play architecture, which means that it can be directly applied to LLMs domains. We are also considering migrating the pipeline to the LLMs domain in future work.

---

### Author Response · Authors · 2024-11-25
**General Response**

We appreciate all the reviewers for taking the valuable time to provide feedback on LIME, which has been very helpful in improving our paper. Since many reviewers raise similar concerns and questions, we will provide a unified response to these issues here, and we encourage all reviewers to read it.

# Discussion about solutions for eliminating data leakage:
**@Reviewer 1M8b and Reviewer Mt3s**

To ‘**Eliminate answer leakage**’, we classify questions into two categories:

'**Text Answerable Questions**' are samples where the answer can be directly inferred from the text of the question and options, without requiring the image.

'**Seen Questions**' refers to samples encountered during the training process. These are easily answerable by MLLMs due to prior exposure.

**We use LLMs to Eliminate Text Answerable Questions**：LLMs are trained on massive amounts of textual corpus, having excellent textual commonsense reasoning ability. In the text-only setting, we use LLMs to filter out text-answerable questions.

**We have eliminated “seen questions” in Semi-Automated Screening Process**:During the Semi-Automated Screening Process, we filter the easy samples, which can be answered correctly by most models (with a correct answer rate greater than 6/9). In fact, this subset of deleted data includes “seen questions” samples.

Additionally, to further support our idea. we test the text-only performance of MLLMs on LIME, As shown in below Table, even the **SOTA** MLLMs(Qwen2-VL-7B ,InternVL-2-8B  ) achieve extremely low scores in the text-only setting, which indicates that our pipeline is robust enough to account for all potential data leakage, and there is no more data leakage in LIME.

| Model                  | AI2D  | ChartQA | COCO-Caption  | InfoVQA | OCRBench   | OK-VQA | POPE  | ScienceQA | TextCaps | TextVQA |
|-|-|-|-|-|-|-|-|-|-|-|
| random                 | 25    | -       | -     | -       | -     | -     | 47    | 41        | -        | -       |
| Qwen2-VL-2B            | 25    | 3.75    | 1.5   | 8.39    | 0.87  | 2.07  | 56.43 | 43.15     | 2.07     | 3.05    |
| Qwen2-VL-7B             | 27    | 5.25    | 2.3   | 9.06    | 0.87  | 5.37  | 44.47 | 41.78     | 3.06     | 4.01    |
| InternVL-2-8B            | 29.3  | 4.42    | 8.19  | 10.51   | 1.09  | 2.98  | 39.28 | 52.74     | 3.88     | 2.96    |
| Xcomposer2-4KHD-7B     | 22.1  | 5.83    | 3.63  | 9.73    | 1.09  | 7.96  | 43.57 | 46.23     | 0.62     | 3.56    |
| LLaVA-1.5-13B          | 22.7  | 3.42    | 7.41  | 7.49    | 1.74  | 7.25  | 40.63 | 33.9      | 3.41     | 3.28    |

# Overall revision of paper:

Based on Reviewer Pra9's suggestion, we have updated the full results of the data size ablation experiment in the appendix A.4.

Based on Reviewer bgCw's suggestion, we replace "model" with "MLLMs" in line 156 to make the expression clearer and add 19 cases related to MMMU and MMBench in the appendix B.

In response to Reviewer Mt3s's feedback, we have refined the description of Section 2.3 on "ELIMINATING ANSWER LEAKAGE" to prevent any potential misunderstandings by the reviewers.

---

### Author Response · Authors · 2024-11-29
**General Response Part(2/2):**

# Motivation of LIME:

LIME is an initial version of a benchmark that embodies two key, enduring motivations:

## 1. Most benchmarks contain low-quality, noisy data:
As mentioned in Figure 2 of our paper, "Most benchmarks contain low-quality, noisy data, which does not accurately reflect the true capabilities of MLLMs." We need a subset of benchmarks from each benchmark that contains a certain amount of data that comprehensively reflects performance across various aspects, and we provide a stable pipeline for selecting this collection.

## 2. Existing benchmarks have large gaps with actual user experience:
 For MLLM evaluation, it is more important to examine what truly relates to the actual user experience rather than just testing the ability to solve tasks simply. However, existing benchmarks have large gaps with actual user experience, and we focus on the parts of existing benchmarks that are most relevant to the real user's needs.


# Contribution of LIME:
## 1. LIME provides  pipeline & guideline for existing benchmarks:
**LIME provides an entirely open-source pipeline**,  which includes three components: **"OPEN-SOURCE MODELS AS JUDGES"**, **"SEMI-AUTOMATED SCREENING PROCESS"**, and **"ELIMINATING ANSWER LEAKAGE"**. By utilizing MLLMs and LLMs, we eliminate data leakage in existing benchmarks, remove potential noise data, and filter out a subset that truly reflects the model's capabilities. Our experimental results show that LIME reduces the cost of benchmark evaluation while maintaining evaluation accuracy, and it better reflects the model's multimodal performance compared to the original benchmark.

**LIME is not only a dataset; but also a universal guideline applicable to benchmarks across all domains:** Although LIME currently selects only 10 benchmarks as the primary subset, it features a plug-and-play architecture that can be applied to any benchmark in any domain. LIME can also serve as a guideline for creating new benchmarks from scratch, enhancing their quality. Additionally, we commit in our paper to continuously update the sub-tasks included in LIME , even if "core" benchmarks emerge in the future, LIME will be able to detect potential noisy data and improve their data quality.
## 2. LIME focuses on the parts of existing benchmarks that are most relevant to the user's needs.

As shown in Figure 4 of our paper, we point out that traditional evaluation metrics(CIDEr) for captioning tasks cannot meet the real user needs, as they only focus on the overlap between the model-generated responses and the ground truth. In addition, LIME achieves over 91% correlation with Wildvision-elo, indicating that LIME, as a collection, is very small and static but has extremely high relevance to user experience—possibly the highest among existing benchmarks.

# Explanations for some other issues:
>  "Once someday, the community will realize there is another benchmark needed for the 'core' MLLM evaluation tasks, then the proposed one will become less meaningful"

In the era of rapid MLLM development, the emergence of new benchmarks every day is an inevitable trend. However, most benchmark creation processes face similar challenges(answer leakage,annotation error). **LIME offers a robust pipeline designed to eliminate potential errors in the data and uncover the most 'core' aspects of a benchmark.** As new benchmarks continue to emerge in the future, we are committed to updating LIME every few months (LIME-v2, LIME-v3, etc.) and providing increasingly valuable test sets.

---

### Author Response · Authors · 2024-11-29
**General Response Part(1/2):**

We would like to sincerely thank all reviewers for their positive recognition of our work. Your feedback is invaluable, not only for improving LIME but also for advancing research in the broader MLLM evaluation field. Below, we provide a comparison of our work with related research and further elaborate on the motivation and contributions of LIME.

**@Reviewer Pra9 and Reviewer 1M8b**

# Systematic Justification
With the rapid development of the MLLMs evaluation field, various benchmarks have emerged. Some works collect data from scratch to create benchmarks, while others filter and process data based on existing benchmarks. **Compared to these works, LIME offers a more robust and general pipeline framework, providing higher-quality benchmarks while maintaining lower production and evaluation costs.**
We use inclusion and exclusion criteria to choose core benchmarks to compare with our LIME:
1. Choose impact benchmarks: filter out papers without 300 citations.
2. Choose general benchmarks: filter out papers to evaluate expert knowledge domains.
3. Choose recent impact publications: add papers on general benchmarks with more than 50 citations released after 2024.

| Bench Name | Task Type      | Production Method           | Answer Leakage Test | Fine-grained Difficulty | Evaluation Cost | Production Cost |
|------------|----------------|-----------------------------|----------------------|-------------------------|-----------------|-----------------|
| **Chartqa**    | Specific Tasks | From Scratch                | ❌                    | ❌              | Middle          | High            |
| **Mmmu**       | General Tasks  | From Scratch                | ❌                    | ✅                   | Middle          | High            |
| **Mmbench**    | General Tasks  | Based on Existing Benchmark | ✅                    |  ❌                | Middle          | Middle          |
| **MMStar**    | General Tasks  | Based on Existing Benchmark | ✅                    | ✅                  | Low             | Middle          |
| **LIME**       | General Tasks  | Based on Existing Benchmark | ✅                    | ✅                   | Low             | Low             |

**Chartqa** [1] collects chart data from four different open-source websites and generates question-answer pairs in the chart domain through a combination of human annotation and language model generation. However, since Chartqa is an early classic dataset, it does not include detailed answer leakage tests or fine-grained difficulty categorization.

**Mmmu**[2] is a large-scale, multidisciplinary, multimodal understanding and reasoning benchmark, which contains 11,500 samples across six different categories. It focuses on evaluating the logic and reasoning capabilities of MLLMs. However, some studies have pointed out that Mmmu heavily relies on pure text-based knowledge and suffers from potential data leakage issues[5,6], failing to adequately assess the multimodal capabilities of MLLMs. In contrast, LIME employs a rigorous data processing pipeline to eliminate potential data leakage issues, providing more challenging tasks for multimodal models.

**Mmbench**[3] is designed to evaluate MLLMs across six different capability dimensions, and it leverages LLMs and MLLMs as judges to filter question sources. However, it only removes text-only and incorrect samples, lacking fine-grained difficulty categorization and still posing risks of potential data leakage. As a result, Mmbench is unable toeffectively distinguish the performance differences between different models. As shown in the Table below, compared to Mmbench, LIME can better reflect the capability differences among various models.

**MMStar**[4] uses LLMs to eliminate data leakage and manually curates a subset of 1,500 data points to mitigate potential data leakage in large multimodal models. However, there are some issues with this approach: 1. It overly relies on manual efforts, and the selection criteria are not fully controllable. 2. The data processing pipeline is not robust enough; it only uses LLMs to address data leakage within LLMs/MLLMs, which may include other potential errors (such as annotation errors, etc.). In comparison, LIME provides a more feasible and generalizable pipeline, which can naturally be extended to other tasks and domains.

[1].Chartqa: A benchmark for question answering about charts with visual and logical reasoning

[2].Mmmu: A massive multi-discipline multimodal understanding and reasoning benchmark for expert agi

[3].Mmbench: Is your multi-modal model an all-around player?

[4].MMStar: Are We on the Right Way for Evaluating Large Vision-Language Models?

[5].Mmmu-Pro: A More Robust Multi-discipline Multimodal Understanding Benchmark

[6].Cambrian-1: A Fully Open, Vision-Centric Exploration of Multimodal LLMs

---

### Author Response · Authors · 2024-12-04
**Summary of reviews, contributions, and changes**

Dear Reviewers and Chairs

We sincerely thank all the reviewers and chairs for their efforts during the rebuttal process. Throughout the discussions, we have received positive feedback and valuable suggestions from the reviewers. We are grateful that they acknowledged our method as novel (Pra9), well-motivated (1M8b, bgCw,Mt3s), and effective（Pra9,1M8b,bgCw) supported by comprehensive experiments (Pra9,1M8b, bgCw, Mt3s). We are also pleased that the subsequent discussions successfully addressed the major concerns raised (Pra9, Mt3s).

Compared to similar related work, the core contribution of LIME is:

1. **LIME provides pipeline & guideline for existing benchmarks**: We have demonstrated that most benchmarks contain low-quality, noisy data, while LIME provides an entirely open-source pipeline. By utilizing MLLMs and LLMs, LIME can eliminate data leakage in existing benchmarks, remove potential noisy data, and filter out a subset that truly reflects the model's capabilities.

2. **LIME focuses on the parts of existing benchmarks that are most relevant to the user's needs.** Existing benchmarks have large gaps with actual user experience，LIME achieves over 91% correlation with Wildvision-elo, indicating that LIME, as a collection, is very small and static but has extremely high relevance to user experience—possibly the highest among existing benchmarks.

Based on the insightful and thoughtful feedback from the reviewers, we have made the following revisions to the paper:

1. Following Reviewer Pra9's suggestion, we have updated the full results of the data size ablation experiment in Appendix A.4 to demonstrate the impact of dataset size on statistical significance.

2. In response to Reviewer bgCw's suggestion, we have replaced "model" with "MLLMs" in line 156 to clarify the expression.

3. We have added a comparison between LIME and related works such as MMMU and MMBench, and included 19 additional cases related to MMMU and MMBench in Appendix B to address the major concerns raised by Reviewer bgCw.

4. In response to Reviewer Mt3s's feedback, we have refined the description in Section 2.3, titled "ELIMINATING ANSWER LEAKAGE," to avoid any potential misunderstandings.

Once again, we sincerely thank you all for your valuable feedback, dedication, engagement, and suggestion; we truly appreciate it.

Best Regards,

Authors

---

### Meta-Review · Area_Chair_6KMZ · 2024-12-22

**Metareview:**

This paper proposed an approach to reduce the unnecessary evaluation samples from the evaluation benchmarks. Concretely, the author proposed to reduce the samples from the benchmarks by 1. employing multiple MLLMs as judge to remove easy samples. 2. employing text only LLM to remove leaked samples (can just answer the questions from the text only) 3. using GPT-4v+ human approach to check the logic and meaning of those questions that all judge models fail to answer.

Strength:
1. The approach is quite comprehensive for reducing the size of the evaluation benchmark.
2. The empirical results are comprehensive.
3. The paper proposed a totally open-sourced approach for combining multiple datasets as a joint benchmark.

Missing from the submission and my major concern:

After reading the paper, the review, and the rebuttal (general responses 1 and 2), I have one major concern for the usefulness of the proposed approach.
1. Evaluation speed: Surely, we can reduce the evaluation time by reducing the benchmark size. However evaluation speed might not be the bottleneck for the whole MLLM pipeline. The training takes much more time than the evaluation. I wonder whether reducing the size of the benchmark is truly meaningful.
2. Human alignment: I have one major concern of reading table 2. I do understand by removing some samples from datasets, the ranking might change. However I wonder whether this is truly reflect the model's capability in T/F Reasoning, VQA, Infographic understanding, OCR, and Science QA? By reading the table 2, GPT-4o achieves way lower performrance than the open sourced model. This is a big claim. I wonder whether this truly represents the GPT-4o is bad on those capability in real-life user experience? If I read this paper correctly, there is no alignment b/w the human preference and the proposed LIME approach. If the answer for the previous question is no (aka GPT-4o might be better on those capability in real-life user experience,) then the significance for this paper might be diminished. Without human alignment, I cannot justify that the LIME score reflect the model's true capability in those domain.

Given those concerns (specifically point 2), I do not think this paper persuaded me for its significance. I would recommend reject.

**Additional Comments On Reviewer Discussion:**

This is a borderline paper (two 5, one 6, and one 8)
The main focus is on those two 5 reviews. For both of the reviews, the major concern is the weakness point 2 I mentioned in the Metareview. After the rebuttal, both reviewers emphasized the concern. However, in the rebuttal, the author didn't fully justify the motivation "Existing benchmarks have large gaps with actual user experience" (general response 2 Motivation section) Without human alignment score, I don't know whether the proposed LIME approach reduce the gaps with actual user experience.

Given this concern, I recommend reject.

---

### Decision · Program_Chairs · 2025-01-22

Reject